# TMPRSS11B promotes an acidified microenvironment and immune suppression in squamous lung cancer

Hari Shankar Sunil [1], Jean R Clemenceau [2], Anthony Grichuk[3], Isabel Barnfather [2], Sumanth R Nakkireddy[2], Luke Izzo [4], Qiang Feng [3], William Hartnett[3], Bret M Evers [5], Lisa Thomas[1], Indhumathy Subramaniyan[6], Li Li[6], William C Putnam [6], Steven Hepensteil [1], Jingfei Zhu[1], Barrett Updegraff[1,7], John D Minna [8,9,10], Ralph J DeBerardinis [11,12,13], Tae Hyun Hwang [2], Jinming Gao [3,8], Trudy G Oliver [4] & Kathryn A O'Donnell [1,8,14 ✉]

## Abstract

**Lung cancer is the leading cause of cancer-related deaths worldwide. Existing therapeutic options have limited efficacy, particularly for lung squamous cell carcinoma (LUSC), underscoring the critical need for the identification of new therapeutic targets. We previously demonstrated that the Transmembrane Serine Protease *TMPRSS11B* promotes the transformation of human bronchial epithelial cells and enhances lactate export from LUSC cells. Here, we evaluate the impact of TMPRSS11B activity on the host immune system and the tumor microenvironment (TME). *Tmprss11b* depletion significantly reduces tumor burden in immunocompetent mice and triggers an infiltration of immune cells. RNA FISH analysis and spatial transcriptomics in the autochthonous *Rosa26-Sox2-Ires-Gfp^LSL/LSL; Nkx2-1^fl/fl; Lkb1^fl/fl* (SNL) model reveal an enrichment of *Tmprss11b* expression in LUSC tumors, specifically in Krt13+ hillock-like cells. Furthermore, utilizing ultra-pH-sensitive nanoparticle imaging and metabolite analysis, we identify regions of acidification, elevated lactate, and enrichment of immunosuppressive (M2-like) macrophages in LUSC tumors. These results demonstrate that TMPRSS11B promotes an acidified and immunosuppressive TME and nominate this enzyme as a therapeutic target in LUSC.**

**Keywords** Squamous Cell Lung Cancer; Transmembrane Serine Protease TMPRSS11B; Lactate-mediated TME Acidification; Immune Suppression; Hillock Cells
**Subject Categories** Cancer; Immunology; Signal Transduction

## Introduction

Lung cancer is the leading cause of cancer-related deaths worldwide (Global Burden of Disease Cancer et al, 2015). Lung cancer is broadly classified into small cell (SCLC) and non-small cell lung cancer (NSCLC), with the latter representing ~85% of all lung cancers. Lung squamous cell carcinoma (LUSC) is one type of NSCLC, accounting for ~30% of all lung cancer cases (Barta et al, 2019; Travis, 2011). These tumors are highly heterogenous and lack targeted therapies. Immune checkpoint inhibitors have emerged as the first-line therapy for LUSC patients, but this approach is effective only in a subset of patients (Haslam and Prasad, 2019; Herbst et al, 2018; Lau et al, 2022; Paik et al, 2019; Santos and Rodriguez, 2022; Satpathy et al, 2021; Stewart et al, 2019). This highlights a critical need for the identification and characterization of new therapeutic targets in LUSC.

Basal cells are thought to be one cell of origin for LUSCs, although accumulating evidence suggests that LUSCs may arise from multiple cell types including club cells and alveolar type II cells (Bairakdar et al, 2025; Ferone et al, 2016; Hanna and Onaitis, 2013; Jeong et al, 2017; Ruiz et al, 2019; Wang et al, 2019; Xu et al, 2014b; Zhang et al, 2017). Hillock cells are a recently described cell type in the lung that were shown to be an injury-resistant reservoir of stem-like cells. They represent a distinctive population of basal stem cells that express Keratin 13 (KRT13) and other genes associated with barrier function, cell adhesion, and immunomodulation. Interestingly, hillock cells were also shown to be one origin of squamous metaplasia, a precursor to LUSC (Deprez et al, 2020; Lin et al, 2024; Montoro et al, 2018; Plasschaert et al, 2018). A better understanding of hillock cell biology and its relationship to LUSC may lead to new therapies for this tumor type.

With the goal of discovering novel genes that drive lung tumorigenesis, we previously performed an unbiased *Sleeping Beauty* (SB)-mediated transposon mutagenesis screen in immortalized human

[1]Department of Molecular Biology, UT Southwestern Medical Center, Dallas, TX, USA. [2]Vanderbilt University Medical Center, Nashville, TN, USA. [3]Department of Biomedical Engineering, UT Southwestern Medical Center, Dallas, TX, USA. [4]Department of Pharmacology and Cancer Biology, Duke University, Durham, NC, USA. [5]Department of Pathology, UT Southwestern Medical Center, Dallas, TX, USA. [6]Texas Tech University Health Sciences Center, Dallas, TX, USA. [7]Pfizer Oncology, Bothell, WA 98021, USA. [8]Harold C. Simmons Comprehensive Cancer Center, UT Southwestern Medical Center, Dallas, TX, USA. [9]Hamon Center for Therapeutic Oncology Research, UT Southwestern Medical Center, Dallas, TX, USA. [10]Department of Pharmacology, UT Southwestern Medical Center, Dallas, TX, USA. [11]Eugene McDermott Center for Human Growth Development, Dallas, TX 75390, USA. [12]Children's Research Institute, Dallas, TX 75235, USA. [13]Howard Hughes Medical Institute, UT Southwestern Medical Center, Dallas, TX 75390, USA. [14]Hamon Center for Regenerative Science and Medicine, UT Southwestern Medical Center, Dallas, TX, USA. ✉E-mail: Kathryn.ODonnell@UTSouthwestern.edu

bronchial epithelial cells (HBECs). This screen revealed that the transmembrane serine protease TMPRSS11B promotes the transformation of HBECs and enhances LUSC tumor growth in immuno-compromised mice (Updegraff et al, 2018). TMPRSS11B belongs to the <u>d</u>ifferentially <u>e</u>xpressed in <u>s</u>quamous cell <u>c</u>ancer (DESC) family of genomically clustered, trypsin-like serine proteases that share commonality in their type-II transmembrane insertion, catalytic triad spacing, and disulfide-bonding, anchoring their serine protease domain to a membrane proximal cysteine (Bugge et al, 2009). We found that *TMPRSS11B* expression is highly upregulated in LUSCs compared to normal lung tissues (Gao et al, 2013). Mechanistic studies and metabolomics further revealed that TMPRSS11B interacts with the lactate monocarboxylate transporter 4 (MCT4) and its obligate chaperone, Basigin (CD147). TMPRSS11B catalytic activity promotes solubilization of Basigin, which enhances MCT4-mediated lactate export, glycolytic flux, and tumor growth, thereby promoting tumorigenesis in LUSC (Updegraff et al, 2018).

Although lactate is usually generated as a by-product of enhanced glycolytic flux in tumor cells, there is significant heterogeneity in its metabolism. Some tumor types prefer the export of lactate through MCT4 to increase glycolysis and drive tumorigenesis. In contrast, other tumor types employ monocarboxylate transporter 1 (MCT1) to import lactate, which can be used for energy production during tumorigenesis (Faubert et al, 2017; Hanahan and Weinberg, 2011; Hong et al, 2016). In addition, extracellular lactate is reported to influence the tumor microenvironment to support tumor growth by inhibiting T-cell function (Dietl et al, 2010; Quinn et al, 2020), recruiting Tregs (Watson et al, 2021), inducing PD-L1 expression (Huang et al, 2024), and by polarizing macrophages to the M2 or immunosuppressive subtype (Bohn et al, 2018; Li et al, 2023; Shan et al, 2020; Zhang and Li, 2020). These studies suggest that lactate metabolism is a vulnerability that may be harnessed for therapeutic targeting of cancer cells.

Tumor-associated macrophages (TAMs) represent a significant fraction of immune cells in the tumor microenvironment and have been shown to promote tumor progression by promoting epithelial-to-mesenchymal transition (EMT), extracellular matrix remodeling through the secretion of proteolytic enzymes, exhaustion and suppression of cytotoxic T cells, and Treg recruitment (Afik et al, 2016; Bahr et al, 2022; Bonde et al, 2012; Maller et al, 2021; Sangaletti et al, 2008). The M2-like subtype is anti-inflammatory, immunosuppressive and tumor promoting, while the M1-like subtype is pro-inflammatory and anti-tumorigenic. In addition to neutrophil infiltration (Kargl et al, 2017), M2-like immunosuppressive macrophages represent a substantial proportion of TAMs in LUSCs, with increased enrichment in the tumor stroma. Moreover, higher M1/M2 ratios correlate with better survival for patients, demonstrating the impact of immunosuppressive TAM subtypes in lung cancer (Hirayama et al, 2012; Jackute et al, 2018; Ma et al, 2010; Sumitomo et al, 2019).

Using an autochthonous model of LUSC coupled with spatial transcriptomics, ultra pH-sensitive nanoparticle imaging, and metabolomics, we show that *Tmprss11b*-expressing lung squamous tumors and the surrounding microenvironment exhibit elevated lactate levels and accumulate immunosuppressive M2-like TAMs. We also demonstrate that *Tmprss11b* is restricted to squamous tumors and is enriched specifically in Krt13⁺ hillock-like cells. Collectively, these studies reveal the establishment of an acidified and immunosuppressive TME in *Tmprss11b*-high LUSC and

suggest a new therapeutic approach of targeting this enzyme in the hillock-like population of squamous lung cancer.

## Results

### *Tmprss11b* depletion reduces tumor growth and enhances CD4 + T cell infiltration

We previously demonstrated that TMPRSS11B inhibition limits tumor growth of human LUSCs in xenograft assays in immunocompromised NOD/SCID Il2rγ$^{-/-}$ (NSG) mice. However, these studies did not fully recapitulate all aspects of tumorigenesis, such as contributions from the tumor microenvironment and the immune system. Based on our previous demonstration that TMPRSS11B promotes lactate export (Updegraff et al, 2018) and the role of lactate in modulating immune cell function, we hypothesized that loss of function of TMPRSS11B would reduce tumor growth and enhance immune cell infiltration in the TME in immunocompetent mice. To test this hypothesis, we used CRISPR/Cas9 to knockout *Tmprss11b* in KLN205 cells, an established syngeneic mouse model of murine lung squamous cell carcinoma (Figs. 1A,B and EV1A) (Kaneko et al, 1980). Cells were transplanted subcutaneously in immunocompetent DBA/2 WT mice, and tumor volumes were assessed. Tumor growth was significantly reduced in *Tmprss11b* knockout tumors compared to control tumors (Figs. 1C and EV1B). We also used doxycycline-inducible short hairpin RNAs (shRNAs) to knockdown *Tmprss11b* in KLN205 cells after tumor initiation (Figs. 1D and EV1C). Cells were transplanted into immunocompetent DBA/2 mice, and mice were maintained on doxycycline water to induce shRNA expression after tumor formation. We observed strong impairment of tumor growth following depletion of *Tmprss11b* (Figs. 1E and EV1D,E). Efficient knockdown of *Tmprss11b* was confirmed in *Tmprss11b*-shRNA expressing tumors (Fig. EV1F). Fluorescent immunohistochemistry (IHC-F) demonstrated that *Tmprss11b* loss of function triggered an accumulation of CD4 + T cells (Fig. 1F), and to a lesser extent CD8 + T cells (Fig. EV1G), in syngeneic tumors. Moreover, we observed a significant decrease in phospho-ERK signaling (Fig. 1G). These findings establish that *Tmprss11b* loss of function alters immune cell infiltration in the TME and suppresses the MAPK signaling pathway.

### *Tmprss11b* expression is enriched in lung squamous tumors

We next investigated the expression of *Tmprss11b* in a panel of tumors derived from autochthonous mouse models of lung cancer representing lung adenocarcinoma (LUAD), lung squamous (LUSC), and small cell lung cancer (SCLC). *Tmprss11b* is highly expressed in LUSC tumors, with the highest expression in the *Rosa26-Sox2-Ires-Gfp$^{LSL/LSL}$;Nkx2-1$^{fl/fl}$;Lkb1$^{fl/fl}$* (SNL) mouse model (Fig. 2A). This model, based on overexpression of the *Sox2* transcription factor and loss of the tumor suppressors *Lkb1* and *Nkx2-1*, accurately recapitulates the histology and microenvironment of LUSCs (Mollaoglu et al, 2018). Interestingly, these mice develop LUSC and mucinous LUAD, thereby providing an ideal setting to investigate intra-tumoral heterogeneity and metabolism.

We initiated tumorigenesis in SNL mice through intratracheal administration of adenovirus expressing Cre (Ad-Cre) (Fig. 2B) and monitored tumor progression over time (Fig. 2C). At 8 months

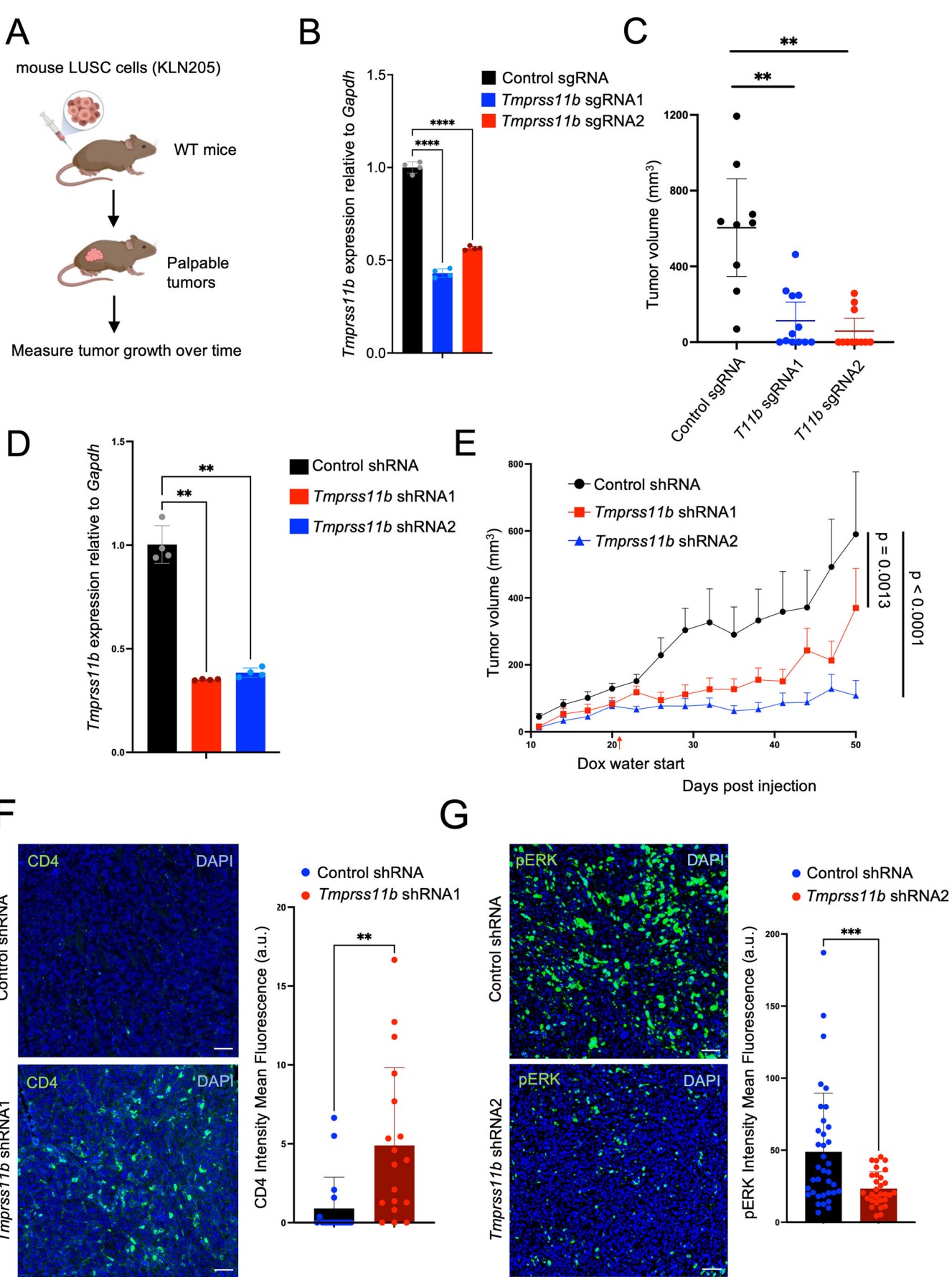

**Figure 1.  *Tmprss11b* depletion reduces tumor growth and enhances immune cell infiltration.**

(A) Schematic of the syngeneic experiments performed using KLN205 murine LUSC cells. Figure created in BioRender. (B) qRT-PCR analysis of *Tmprss11b* mRNA in KLN205 cells expressing control sgRNA or two independent *Tmprss11b* sgRNA. Ordinary one-way ANOVA with Dunnett's multiple comparisons test was used for the analysis. Plot represents mean ± standard deviation (SD); $n = 4$ per group (technical replicates), ****$P < 0.0001$ for all comparisons. The experiment was repeated three times for confirmation (biological replicates). (C) Quantification of tumor volumes of KLN205 tumors expressing control or *Tmprss11b* sgRNA on day 45 (terminal measurement) post injection in syngeneic DBA/2 wild-type mice ($n = 9$ control sgRNA mice; $n = 12$ *Tmprss11b* sg1 mice; $n = 12$ *Tmprss11b* sg2 mice) (biological replicates), $P = 0.0036$ (*T11b* sgRNA1), $P = 0.0022$ (*T11b* sgRNA2). Brown–Forsythe and Welch ANOVA test with Dunnett's T3 multiple comparisons test was used for the statistical analysis. The plot represents mean with 95% confidence interval (CI). (D) qRT-PCR analysis of *Tmprss11b* mRNA in KLN205 cells expressing doxycycline-inducible control shRNA or two independent shRNA sequences targeting *Tmprss11b*. RM one-way ANOVA with Dunnett's multiple comparisons test was used for the analysis. Plot represents mean ± standard deviation (SD); $n = 4$ per group (technical replicates), $P = 0.0012$ (*T11b* shRNA1), $P = 0.0013$ (*T11b* shRNA2). The experiment was repeated three times for confirmation (biological replicates). (E) Quantification of tumor volumes of KLN205 cells expressing doxycycline-inducible control or *Tmprss11b* shRNA in syngeneic DBA/2 wild-type mice. ($n = 9$ control shRNA mice; $n = 6$ *Tmprss11b* sh1 mice; $n = 8$ *Tmprss11b* sh2 mice, biological replicates), $P = 0.0013$ (*T11b* shRNA1), $P < 0.0001$ (*T11b* shRNA2). Linear mixed-effects models were used to investigate the differences in tumor volume over time among the three groups. The plot represents the standard error of the mean (SEM). (F) Fluorescent immunohistochemistry (IHC-F) staining for CD4 + T cells in KLN205 tumor sections, with quantification. An unpaired *t* test with Welch's correction was used to compare CD4 + T cells between the groups ($n = 6$ fields per tumor section, three tumors per group, biological replicates), $P = 0.0042$. Plot represents mean ± SD. Scale bar, 50 µm. (G) Fluorescent immunohistochemistry (IHC-F) staining for phosphorylated ERK (pERK) in KLN205 tumors, with quantification (right). An unpaired *t* test with Welch's correction was used to compare phosphorylated ERK levels between the groups ($n = 10$–14 fields per tumor section, three tumors per group, biological replicates), $P = 0.0008$. Plot represents mean ± SD. Scale bar, 50 µm. Source data are available online for this figure.

post-infection, we observed appreciable lung tumor burden and distinct regions of squamous tumors (LUSC) and mucinous adenocarcinomas (LUAD) identified by H&E staining (Fig. 2D). We next assessed the spatial distribution of *Tmprss11b* expression with RNA fluorescent in situ hybridization (FISH) (RNAscope). We observed selective enrichment of *Tmprss11b* in squamous lung tumors but not in mucinous adenocarcinomas or normal lungs (Fig. 2E–G). In contrast, *Sox2*, the oncogenic driver in this model, was expressed in LUSC and LUAD tumors (Fig. 2G). These data demonstrate that *Tmprss11b* is selectively expressed in murine lung squamous tumors.

### *Tmprss11b* expression correlates with an increase in squamous markers and oncogenic signaling pathways

Given the heterogeneity observed in SNL lung tumors, we next investigated the gene expression pathways that associate with *Tmprss11b* expression using spatial transcriptomics. High quality sequencing reads were obtained with the Visium CytAssist (10x Genomics) platform (Fig. EV2A) and analysis revealed multiple clusters, indicating spatial heterogeneity (Fig. 3A). We annotated the sections as LUSC and LUAD based on histology for additional analyses, including 3 regions as LUSC (S1, S2, S3) and 2 regions as LUAD (A1, A2) (Fig. 3B). We assessed the distribution of *Tmprss11b* transcripts and observed unique enrichment in the LUSCs, validating our RNA-FISH results (Fig. 3C). As expected, the squamous marker *Trp63* and the mucinous adenocarcinoma marker *Hnf4a* exhibited enrichment in LUSCs and LUADs, respectively (Fig. 3C).

We next quantified *Tmprss11b* expression in all annotated regions. Although all LUSC regions expressed higher levels of *Tmprss11b* compared to LUAD regions, the S1 and S2 regions exhibited ~fourfold higher levels of expression compared to the S3 region (Fig. EV2B,C). To elucidate pathways elevated in squamous tumors with high *Tmprss11b* expression, we performed gene set enrichment analysis (GSEA) on *Tmprss11b*-high (S1, S2) versus *Tmprss11b*-low (S3) LUSC regions and observed an enrichment of oncogenic and immune cell signaling pathways (Fig. 3D). In addition, we performed differential gene expression (DEG) analysis on *Tmprss11b*-high versus low regions and observed an increase in

genes known to be elevated in squamous tumors (*Krt16* and *Sprr2d*) and genes that promote tumorigenesis, including *Tgm3*, *Krt16*, and *S100a7a* (Fig. 3E) (Arora et al, 2023; Ferone et al, 2016; Kwon et al, 2023; Liu et al, 2022a). We also performed GSEA and DEG comparing *Tmprss11b*-high LUSCs (S1, S2) versus LUADs (A1, A2) and observed a similar enrichment of genes and pathways (Fig. EV2D,E). Immunohistochemistry (IHC) analysis of the LUSC markers KRT16 and LYPD3 demonstrated increased expression in LUSCs compared to LUADs, validating our spatial transcriptomics data (Fig. EV2F). These findings demonstrate that tumors expressing high levels of *Tmprss11b* exhibit an enrichment for squamous markers and an oncogenic gene expression signature.

### TMPRSS11B is enriched in the hillock cell population and induced by KLF4

Given the robust expression of *Tmprss11b* in squamous tumors and the strong correlation with squamous markers and oncogenic signaling pathways, we hypothesized that TMPRSS11B may be expressed in the cell of origin in LUSCs. Basal cells are hypothesized to be the precursor of squamous tumors (Hanna and Onaitis, 2013; Jeong et al, 2017), and this is supported by machine-learning-based approaches in human tumors (Bairakdar et al, 2025). However, lineage tracing studies have demonstrated that squamous tumors can arise from multiple cell types in the lung, including club cells and alveolar type II (AT2) cells (Ferone et al, 2016; Ruiz et al, 2019; Wang et al, 2019; Xu et al, 2014b; Zhang et al, 2017). Recent findings have identified a new cell type in the pseudostratified lung epithelium known as hillock cells that can give rise to squamous metaplasia, a precursor of squamous tumors (Lin et al, 2024). These are collections of multilayered injury-resistant cells that express KRT13 and consist of keratinized upper layers of luminal cells that protect underlying layers of KRT13⁺, TP63⁺ basal stem cells (Deprez et al, 2020; Hewitt and Lloyd, 2021; Lin et al, 2024; Montoro et al, 2018; Plasschaert et al, 2018). Interestingly, these domains harbor a unique population of basal stem cells that express genes associated with barrier function, cell adhesion, and immunomodulation (Bae et al, 2022; Dragomir et al, 2012; Hewitt and Lloyd, 2021; Kumar et al, 2015; Montoro et al, 2018; Nakamura et al, 2019; Ng et al, 2011). Moreover, there is a

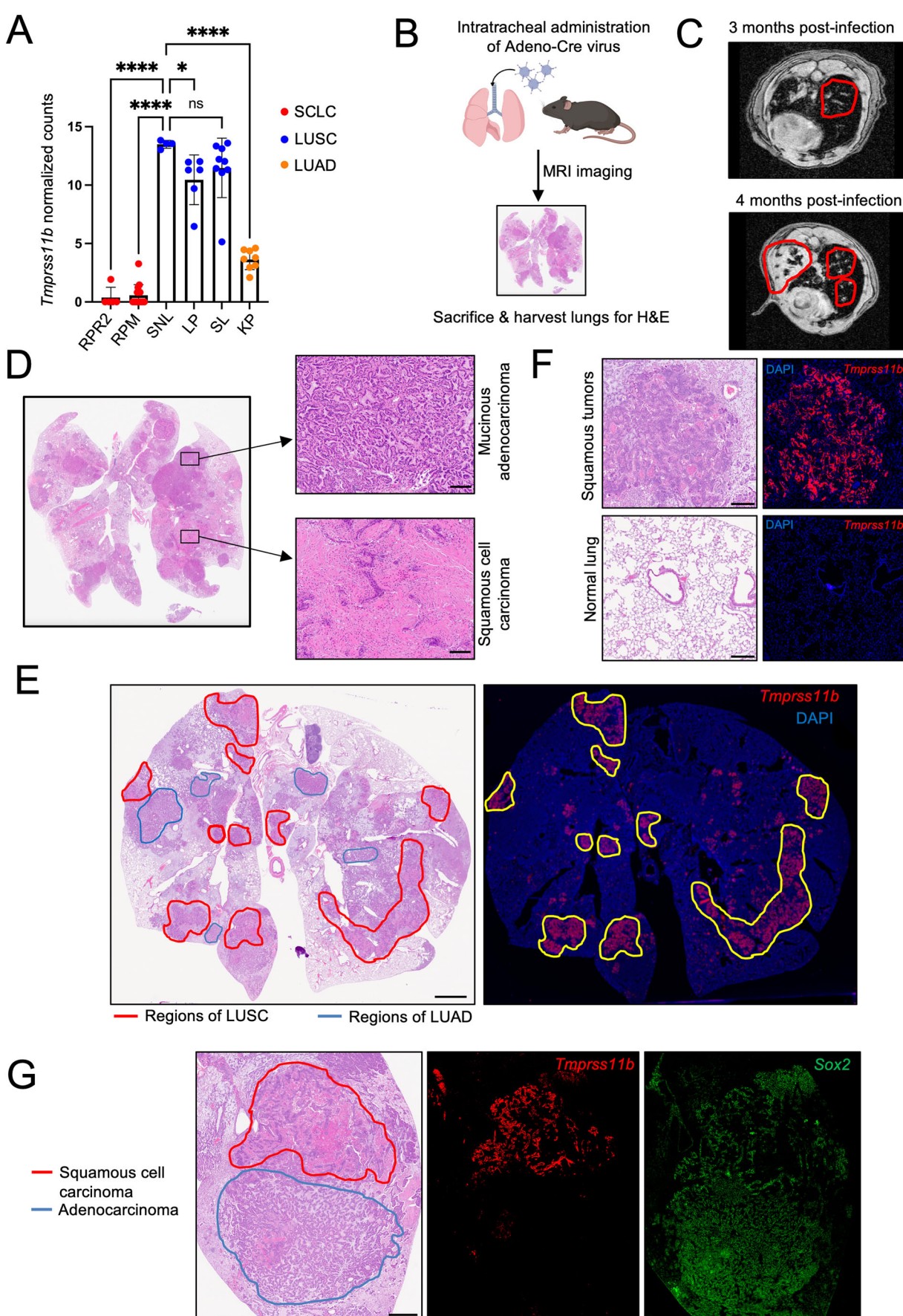

**Figure 2.** *Tmprss11b* expression is enriched in mouse lung squamous tumors.

(A) RNA sequencing analysis of tumors from various mouse models of lung cancer. The y-axis represents the normalized counts for *Tmprss11b*. One-way ANOVA with Dunnett's multiple comparisons test was used for the statistical analysis (RPR2 mice $n = 5$; RPM mice $n = 15$; SNL mice $n = 4$; LP mice $n = 6$; SL mice $n = 9$; KP mice $n = 8$, biological replicates), ****$P < 0.0001$ (RPR2, RPM, KP), *$P = 0.0133$ (LP). Plot represents mean ± SD. (B) Schematic representation of Ad-Cre mediated tumor induction in SNL mice. Figure created in BioRender. (C) Representative MRI images of the mice in (B), 3- & 4-months post infection. Red outlines denote tumors (Biological replicates $n > 3$). (D) Representative H&E images of SNL mouse lung, 7 months post infection with Ad-Cre showing distinct regions of LUSC and mucinous LUAD (Biological replicates $n > 3$). Scale bar, 100 µm. (E) H&E image of SNL mouse lung (11 months post infection with Ad-Cre) and RNAscope of *Tmprss11b* on a serial section. Left, red outline denotes squamous tumors based on H&E staining. Right, yellow outline denotes regions with *Tmprss11b* expression (red) corresponding to the regions of squamous tumors. The staining was repeated three times with serial sections (technical replicates) and with lung sections from different mice ($n = 4$, biological replicates). Scale bar, 2 mm. (F) Zoom-in of (E) showing *Tmprss11b* expression by RNAScope in squamous tumors (top panel) and normal lung (bottom panel). Scale bar, 200 µm. (G) Representative H&E image with annotations and RNAscope analysis of *Tmprss11b* (red) and *Sox2* (green) in SNL lung sections. Scale bar, 400 µm. Source data are available online for this figure.

growing appreciation that membrane-anchored serine proteases are important regulators of epithelial development and barrier function (Szabo and Bugge, 2020b). RNA sequencing of mouse syngeneic tumors revealed that keratin genes were downregulated in *Tmprss11b*-depleted cells (Fig. EV3A). Conversely, DEG analysis of *Tmprss11b*-high squamous tumors and LUSC patient data from TCGA uncovered an enrichment of several keratin genes, including *KRT13*, *KRT6*, *KRT10*, and *KRT14* (Fig. EV3B–E). Given these observations, we sought to determine if TMPRSS11B is expressed in KRT13[+] hillock cells.

Findings from a recent study identified a hillock-like cell population in human and mouse lung squamous cell carcinomas (Izzo et al, 2025). Consistent with this observation, we find that *Tmprss11b* is co-expressed with *Keratin 13* (*Krt13)* and *Keratin 6a* (*Krt6a*) in mouse LUSC tumors (Fig. 4A). Using hillock, basal, and mucinous gene signatures, we assigned regions of enrichment for these signatures in our spatial transcriptomics dataset. We find that the hillock and basal gene signatures were highly enriched in the *Tmprss11b*-high squamous tumors, while the mucinous gene signature was enriched in the adenocarcinomas (Fig. EV4A,B). We performed RNA-FISH analysis of *Tmprss11b*, *Trp63*, and *Krt13* in SNL tumors and observed an enrichment of *Trp63* in the basal layer and expression of *Krt13* in the suprabasal layer in squamous tumors, consistent with a recent study (Lin et al, 2024) (Fig. 4B). Interestingly, *Tmprss11b* expression was suprabasal to *Trp63* and co-expressed with *Krt13*.

To extend these findings to human lung squamous tumors, we analyzed LUSC RNA-sequencing data from TCGA. Indeed, TMPRSS11B was upregulated in human LUSCs compared to LUAD (Fig. 4C). A detailed analysis of cell clusters using a hillock gene signature found that a *TMPRSS11B*-high cluster also expresses *KRT13* (Izzo et al, 2025). Importantly, this study demonstrated that the Krüppel-like factor 4 (KLF4) transcription factor induces a hillock-like gene signature in human bronchial epithelial cells (BEAS-2B) (Izzo et al, 2025). Given the importance KLF4 in regulating a hillock-like cell state and its regulatory role in a variety of cancers (He et al, 2015; Hu et al, 2015; Riverso et al, 2017; Yan et al, 2016), we sought to determine if TMPRSS11B is induced by KLF4. We performed quantitative RT-PCR for *TMPRSS11B* on RNA isolated from BEAS-2B cells following doxycycline-inducible expression of green fluorescent protein (GFP) control or KLF4. Indeed, *TMPRSS11B* was induced by ~40-fold in KLF4-expressing BEAS-2B cells (Fig. 4D). Overall, these data show that *TMPRSS11B* is induced by KLF4 and is expressed in the KRT13+ hillock-like cell population in lung squamous tumors.

## *Tmprss11b*-high squamous tumors exhibit infiltration of M2-like macrophages

Given the enrichment of squamous markers and oncogenic signaling in *Tmprss11b*-expressing squamous tumors (Figs. 3D and EV2D), and the known link between lactate metabolism and immune regulation, we next examined the immune cell populations that infiltrate LUSC tumors. Previous studies have reported higher levels of neutrophil infiltration in LUSCs compared to LUADs (Kargl et al, 2017; Salcher et al, 2022). We performed GSEA and observed an enrichment for pathways associated with neutrophils and macrophages in the *Tmprss11b*-high LUSCs (Fig. EV5A). Known markers of neutrophil infiltration including *Cd11b* (*Itgam*), *Cxcl3*, and *Cxcl5* were expressed in LUSCs (Fig. EV5B). IHC analysis confirmed higher levels of several neutrophil markers including MPO, ITGAM (CD11b) and LY6G (Fig. EV5C), consistent with prior studies in this mouse model (Mollaoglu et al, 2018). Moreover, cell deconvolution analysis identified an enrichment of additional immune cells subsets, including classical monocytes, alveolar macrophages and leukocytes, with high expression of *Tmprss11b* (Figs. 5A,B and EV6A,B). GSEA of *Tmprss11b*-high (S1, S2) vs. *Tmprss11b*-low LUSC (S3) and *Tmprss11b*-high LUSC (S1, S2) vs. LUAD (A1, A2) revealed an enrichment for similar pathways (Figs. 5C and EV6C). Given the presence of opposing subtypes of macrophages observed in the tumor microenvironment, based on their ability to promote or suppress tumor progression, we assessed macrophage subtypes that were enriched with higher levels of *Tmprss11b*. DEG analysis of the spatial data identified an enrichment of common macrophage markers such as *Cd68*, and additional genes typically associated with immunosuppressive TAMs or M2-like macrophages including *Arg1*, *Hmox1*, *Msr1*, *Trem2*, *Spp1*, and *Ctsk* (Fig. 5D and EV6D) (Hirayama et al, 2012; Katzenelenbogen et al, 2020; Liu et al, 2022b; Ma et al, 2022a; Muliaditan et al, 2018; Wang et al, 2024; Wu et al, 2022; Yanai et al, 2021; Yu et al, 2019). To validate this, we performed IHC and observed higher levels of MSR1 (CD204), HMOX1, and ARG1 expression in LUSCs compared to LUADs (Fig. 5E). In contrast, the oncogenic driver in this mouse model, SOX2, was expressed in both LUSCs and LUADs. These findings demonstrate that *Tmprss11b* expression correlates with infiltration of immunosuppressive (M2-like) macrophages.

## *Tmprss11b*-high squamous tumors and acidified regions of the TME are enriched for immunosuppressive macrophages

Given the role of TMPRSS11B in promoting lactate export and the ability of lactate to promote polarization of macrophages to the

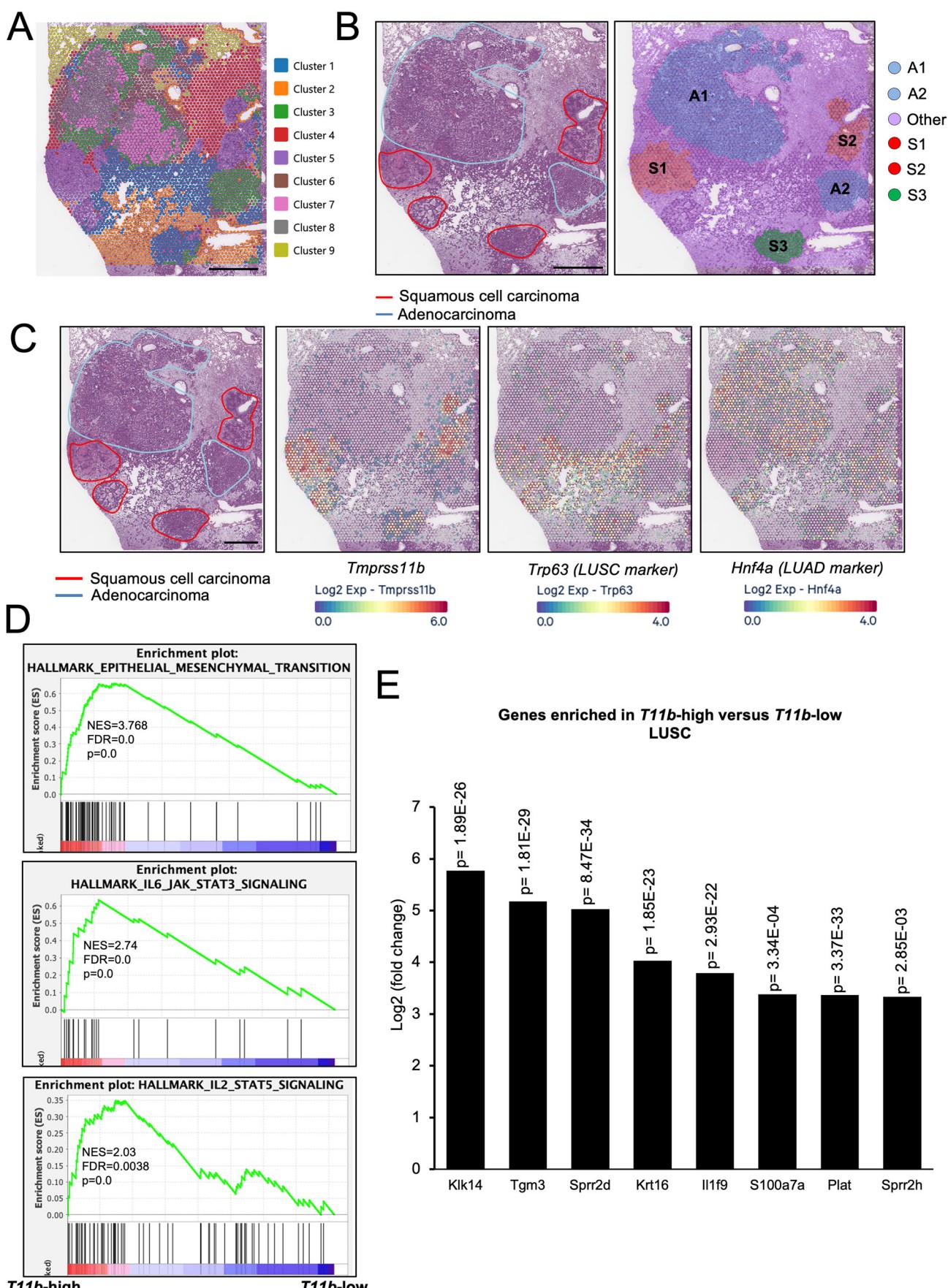

**Figure 3.** *Tmprss11b* expression correlates with an increase in squamous markers and oncogenic signaling pathways.

(A) Image depicting the different clusters identified in a SNL lung section based on spatial transcriptomic analysis. Scale bar, 2 mm. (B) H&E image of the lung section from A) (left) and manual annotation of different regions (right); LUSC (S1, S2, S3) and LUAD (A1 and A2). Scale bar, 2 mm. (C) H&E image of the lung section from A), annotated with regions of LUSC and LUAD, and corresponding spatial plots depicting the distribution of the indicated mRNAs. Scale bar, 1 mm. (D) Gene set enrichment analysis (GSEA) from *T11b*-high versus low squamous tumors (S1&S2 versus S3) with normalized enrichment scores (NES), false discovery rate (FDR) and P values for the indicated gene signatures. The nominal P and FDR values were obtained from the "GSEA Preranked" tool (from Broad Institute) using a weighted scoring scheme. Gene sets were evaluated based on the default normalized enrichment score method, and statistical significance was determined by bootstrapping with 1000 permutations. (E) Bar graph representing top squamous genes/oncogenes from differential gene expression (DEG) analysis of *Tmprss11b*-high versus low LUSC spatial transcriptomics data. Differential gene expression was calculated using Seurat's FindAllMarkers function, and direct comparisons between two classes and corresponding P values were obtained using Seurat's FindMarkers function with the Wilcoxon rank-sum test with Bonferroni P value correction. Source data are available online for this figure.

M2-subtype (Bohn et al, 2018; Li et al, 2023; Shan et al, 2020; Zhang and Li, 2020), we hypothesized that the infiltration of M2-like macrophages in *Tmprss11b*-high squamous tumors was associated with elevated levels of lactate in the tumor microenvironment. To investigate this, we utilized ultra pH-sensitive (UPS) nanoparticles conjugated to the indocyanine green (ICG) fluorophore (Appendix Fig. S1A) (Feng et al, 2019). We utilized a UPS nanoparticle with a threshold pH of 5.3 that disassembles and electrostatically binds to the surrounding tissue when the pH is below 5.3 (Feng et al, 2024). The illuminated ICG signal defines the regions of increased acidity that are driven by the secretion of lactic acid. We administered the UPS nanoparticles to control animals (without Cre) and tumor-bearing SNL mice (with Cre) by tail vein injection and performed ex vivo imaging of the lungs (Appendix Fig. S1B). We observed a high ICG signal in the lungs of tumor-bearing SNL mice administered with Ad-Cre, indicating increased acidification (Fig. 6A,B). Moreover, the ICG signal accumulates in tumor stromal regions with higher vascular density and infiltrative macrophages expressing immunosuppressive markers, including *Cd68* and *Spp1* (Bill et al, 2023) (Fig. 6C). Consistent with this, our prior data showed the highest level of acidity at the tumor and stromal interface, where cancer cells secrete lactic acid into the stromal areas, leading to the nanoprobe activation and internalization by stromal cells (Feng et al, 2024). We performed GSEA and observed an enrichment of pathways associated with macrophages and monocytes, and oncogenic signaling in regions with low pH (Fig. 6D; Appendix Fig. S1C). Cell deconvolution analysis further revealed an enrichment for select immune cell subtypes, including monocytes and macrophages, in low-pH regions (Appendix Fig. S1D). Finally, DEG analysis and IHC validation revealed the expression of M2-like and TAM gene signatures (Fig. 6E), and expression of HMOX1 and MSR1 in regions with high UPS signal, respectively (Fig. 6F). Taken together, these findings reveal the presence of acidified regions in lung squamous tumors that accumulate immunosuppressive M2-like macrophages.

### *Tmprss11b*-high squamous tumors and low pH regions have elevated levels of lactate

Given the observed enrichment of immunosuppressive M2-like macrophages in *Tmprss11b*-high squamous tumors and the surrounding TME, we reasoned that higher levels of lactate should be present in the acidified regions. To demonstrate that regions with low pH and increased acidification in *Tmprss11b*-expressing tumors accumulate lactate, we performed laser capture micro-dissection (LMD) and mass spectrometry to quantify metabolites in the lungs from SNL mice (Fig. 7A). We used H&E staining to

annotate regions of LUSC, LUAD, and normal lung. The ICG signal was used to annotate regions of low pH (Fig. 7B; Appendix Fig. S2A). Quantification of metabolite levels revealed a significant enrichment of lactate in LUSC compared to LUAD and surrounding normal lung tissues (Fig. 7C). Consistent with this, the low pH regions overlap with tumors to a greater extent than the high pH regions (Appendix Fig. S2B). Moreover, we observed elevated lactate levels in low pH regions identified with the ultra pH-sensitive nanoprobe (Fig. 7D). Finally, we identified significant enrichment of glycolytic gene signatures but not TCA cycle signatures, and increased expression of the glycolysis-associated transporters GLUT1 and MCT4 in *Tmprss11b*-high squamous tumors (Appendix Fig. S3A–C). These findings agree with the observation that squamous tumors have increased glycolytic flux compared to adenocarcinomas (Enfield et al, 2024; Goodwin et al, 2017). Collectively, our data demonstrate that LUSC tumors acidify the TME due to their high expression of *Tmprss11b*, resulting in an immunosuppressive environment that enhances tumor growth.

## Discussion

Based on our prior studies demonstrating a role for TMPRSS11B in promoting tumorigenesis and enhancing lactate export in vitro, we hypothesized that TMPRSS11B expression would promote an immunosuppressive tumor microenvironment in vivo. Indeed, loss of TMPRSS11B in a syngeneic mouse model confirmed the importance of TMPRSS11B in LUSC tumorigenesis and identified alterations in immune cell populations and signaling upon depletion of *Tmprss11b*. Furthermore, using the *Rosa26-Sox2-Ires-Gfp^{LSL/LSL};Nkx2-1^{fl/fl};Lkb1^{fl/fl}* (SNL) GEMM model of LUSC, we observed specific expression of TMPRSS11B in lung squamous tumors. A similar expression pattern is observed in the LP (*Lkb1^{fl/fl}, Pten^{fl/fl}*) GEMM model, where *Tmprss11b* is one of the top upregulated genes in lung squamous tumors (Xu et al, 2014a). Leveraging the SNL model, we performed spatial transcriptomics on lung tumors, which revealed increased oncogenic signaling and accumulation of immunosuppressive M2-like macrophages in *Tmprss11b*-expressing squamous tumors. We assessed the in vivo acidification of squamous tumors using ultra pH-sensitive nanoparticles and uncovered the presence of low pH regions adjacent to tumors that are rich in immunosuppressive M2-like macrophages. Given the ability of lactate to promote macrophage polarization and immunosuppression (Bohn et al, 2018; Li et al, 2023; Shan et al, 2020; Zhang and Li, 2020), we hypothesized that lactate is present in these regions. Accordingly, laser capture microdissection followed by mass spectrometry demonstrated that regions with low pH exhibited elevated lactate abundance. Taken together, these results

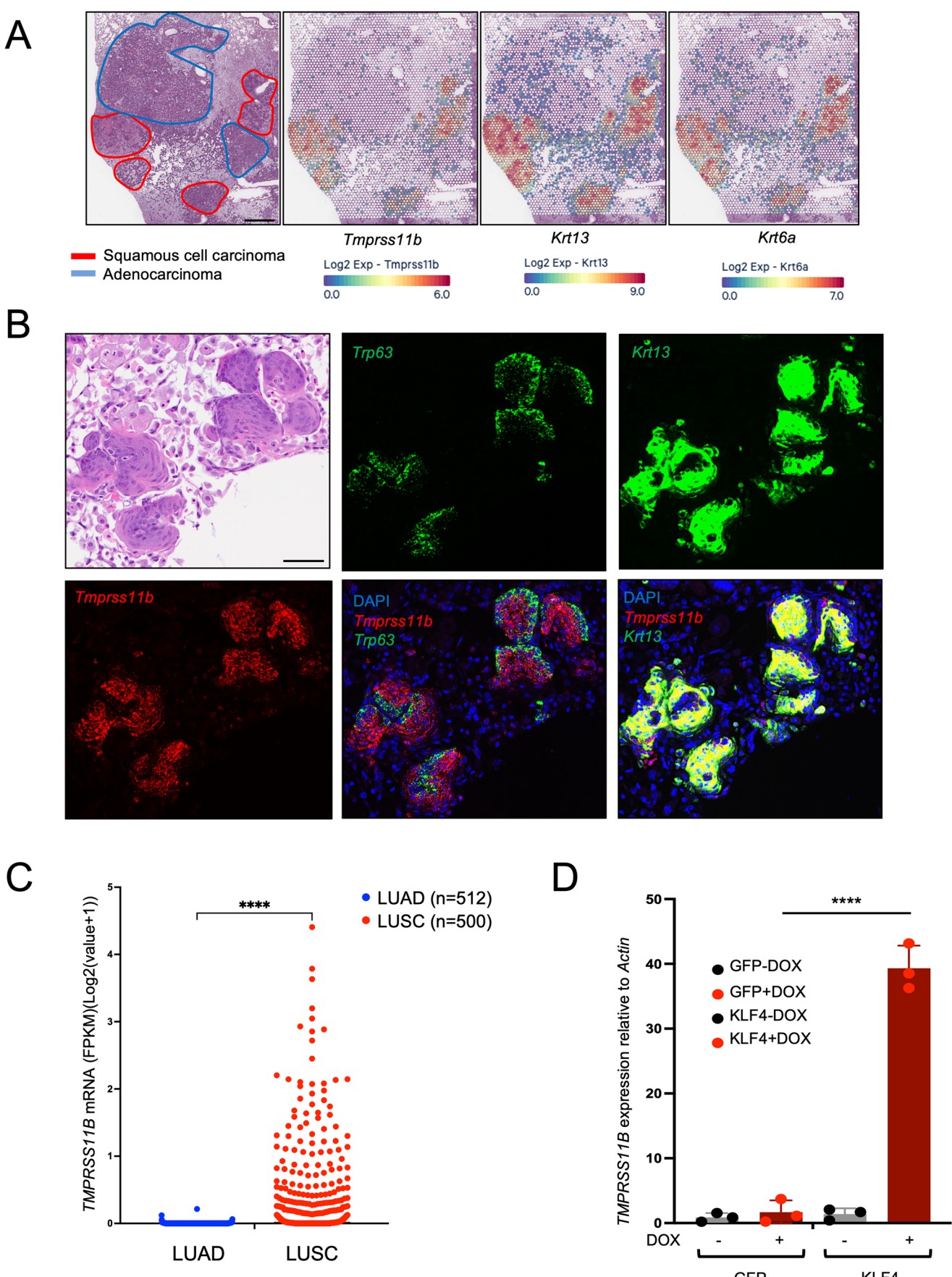

**Figure 4. TMPRSS11B is enriched in the KRT13$^+$ hillock-like cells and induced by KLF4.**

(A) H&E image of the lung section from 3 A), annotated with regions of LUSC and LUAD, (left) and corresponding spatial plots depicting the distribution of the indicated mRNAs. Scale bar, 1 mm. (B) Representative H&E image and RNAscope analysis of *Tmprss11b*, *Trp63*, and *Krt13* in SNL lung sections. The staining was repeated with lung sections from different mice (*n* = 2–3, biological replicates). Scale bar, 50 μm. (C) *Tmprss11b* transcript abundance in LUSC patient tumors (*n* = 500) relative to LUAD tumors (*n* = 512) (data obtained from TCGA-GDC (Cancer Genome Atlas Research et al, 2013; Heath et al, 2021)). Welch's *t* test was used for the analysis, ****$P$ < 0.0001. Plot represents median with 95% CI. (D) qRT-PCR analysis of *Tmprss11b* mRNA in BEAS-2B cells expressing doxycycline-inducible GFP or KLF4. Ordinary one-way ANOVA with Dunnett's multiple comparisons test was used for the analysis. Plot represents mean ± SD; *n* = 3 (technical replicates) per group, ****$P$ < 0.0001. The experiment was repeated two times to confirm the observations. Source data are available online for this figure.

support a model whereby TMPRSS11B expression in LUSC results in acidification of the TME due to enhanced lactate export. This produces an immunosuppressive environment, characterized by the infiltration of M2-like macrophages, which promotes tumor progression (Fig. 7E).

Treatment modalities for LUSC patients have significantly advanced in recent years. Immunotherapy is approved as a first-line therapy and has shown promising improvement in progression-free survival, either as a monotherapy or in combination with chemotherapy. However, only a subset of the patients respond with durable benefit and patients often develop toxicity or resistance to immunotherapy, warranting the need for alternative therapies (Lau et al, 2022; Paik et al, 2019; Paz-Ares et al, 2018; Reck et al, 2016; Stewart et al, 2019). In addition, efforts to target frequently upregulated molecules in LUSC such as FGFR, PI3K and CDK4/6 have been largely unsuccessful (Paik et al, 2019). Collectively, these data suggest the need for identification and development of new targeted therapies for molecules specifically upregulated in LUSC. The selective upregulation of TMPRSS11B in lung squamous tumors and the impact on immune suppression and oncogenic signaling pathways nominate this cell surface enzyme as a promising therapeutic target.

Our studies suggest that the enrichment of TMPRSS11B in lung squamous tumors may be attributed to its expression in hillock cells. The unique properties of this cell type include their ability to form stratified structures with a gradient of KRT13 and TP63 expression. This results in upper layers of ciliated luminal (termed hillock luminal) cells marked by expression of KRT13 and underlying layers of basal stem (termed hillock basal) cells marked by high expression of TP63 (Hewitt and Lloyd, 2021; Lin et al, 2024; Montoro et al, 2018). The hillock basal cells appear to be proliferative and express squamous differentiation genes at low levels, while the hillock luminal cells consist of tightly interlocked squamous cells that form a protective barrier for the underlying stem cells. Hillock cells are highly resistant to insults, including chemical injury and viral infection (Lin et al, 2024). Furthermore, hillock cells express genes involved in cellular adhesion and immunomodulation, including *Cldn3*, *S100a11*, *Ecm1*, *Lgals3*, and *Anxa1* (Bae et al, 2022; Dragomir et al, 2012; Hewitt and Lloyd, 2021; Kumar et al, 2015; Montoro et al, 2018; Nakamura et al, 2019; Ng et al, 2011). Interestingly, recent studies have suggested an important role for members of the type-II transmembrane serine protease (TTSP) family in barrier function (Szabo and Bugge, 2020a). Given the unique enrichment of TMPRSS11B in hillock-like cells, it is plausible that TMPRSS11B contributes to barrier function and immune regulation mediated by hillock cells.

A recent study identified hillock cells as one origin of squamous metaplasia, a precursor of LUSC (Lin et al, 2024). Another study suggested that KLF4 can induce hillock gene signatures in human

bronchial epithelial cells (Izzo et al, 2025). This latest work (Izzo et al, 2025), taken together with our findings that KLF4 induces *Tmprss11b*, suggests a model whereby KLF4 is transiently upregulated in hillock basal cells, giving rise to suprabasal hillock luminal cells that express KRT13 and TMPRSS11B. This population of cells is hypothesized to function as a barrier, protecting the underlying hillock basal stem cells from various insults. The role of hillock cells and the hillock-like state in the initiation and progression of squamous tumors, and the role of TMPRSS11B in this process, is not completely understood. Future studies are warranted to fully elucidate the role of TMPRSS11B in the hillock-like cancer cell state and to identify substrates of this serine protease in normal lung development and in LUSC.

Macrophages comprise a significant fraction of tumor-associated immune cells in the lung and have been shown to influence tumor growth and treatment outcomes (Cassetta and Pollard, 2018; Conway et al, 2016; Ruffell and Coussens, 2015). In addition, the presence of the pro-inflammatory M1 and anti-inflammatory M2 subtypes adds an additional layer of complexity to understanding the impact of macrophages in the TME. Recent advances from single-cell RNA sequencing and spatial transcriptomic studies of patient tumors have uncovered the presence of a continuum of TAM subtypes in the TME, beyond the M1 and M2 classes. These TAMs are characterized by their specific gene expression profiles, whose combined functional outcomes, along with the cancer type, dictate their influence on the TME. It is also important to note that these diverse TAM subtypes can co-exist in the same TME, and their effects, whether synergistic or antagonistic, determine their impact on the TME (Hochstadt et al, 2025; Ma et al, 2022b; Mantovani et al, 2024; Yang et al, 2024). We acknowledge this continuum and denote the immunosuppressive TAMs "M2-like" in this study. Indeed, higher levels of M2-like macrophages are associated with a decrease in overall survival (OS) of LUSC patients (Sumitomo et al, 2023; Wu et al, 2016). Our studies have uncovered an enrichment of M2-like macrophages in *Tmprss11b*-high squamous tumors and the surrounding acidified regions. In addition, we identified higher lactate levels in these regions. Taken together with our previously identified role for TMPRSS11B in promoting lactate export and the reported role of lactate in polarization of macrophages to the M2 subtype, these results suggest an association of TMPRSS11B with the recruitment or production of immunosuppressive M2-like macrophages. Recent findings have also implicated a role for IL6-JAK-STAT3 signaling in promoting polarization of macrophages to the M2 subtype (Jiang et al, 2019). Interestingly, our studies uncovered upregulation of an 'IL6_JAK_STAT3_Signaling' pathway signature associated with increased *Tmprss11b* expression, further hinting at a potential role for TMPRSS11B in promoting macrophage polarization.

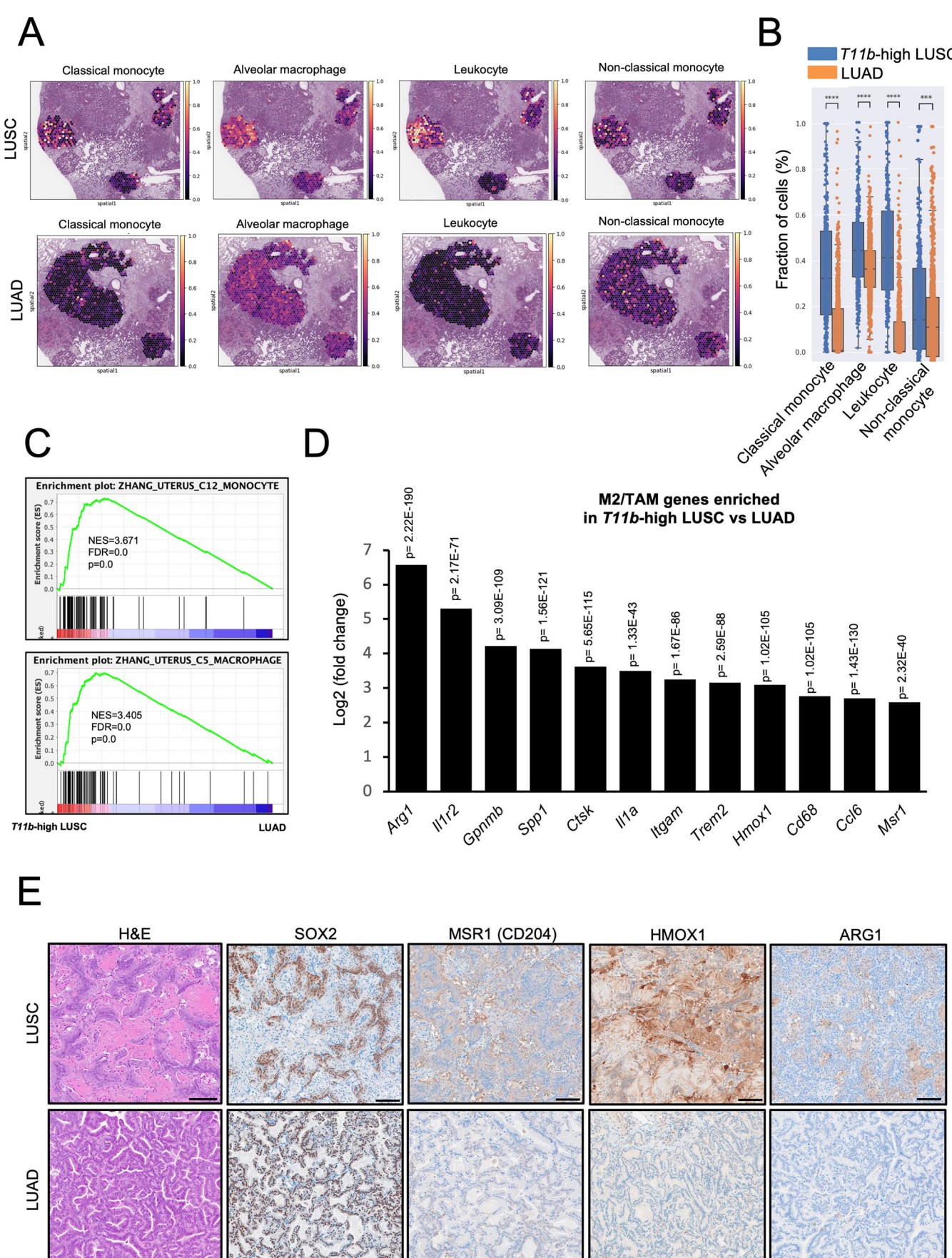

◄ **Figure 5. Infiltration of M2-like macrophages in *Tmprss11b*-high squamous tumors.**

(A) Spatial plots from cell deconvolution analysis showing the distribution of various immune cell populations in LUSCs (top) and mucinous LUADs (bottom). (B) Quantification of immune cell populations in (A). A two-sided Mann–Whitney–Wilcoxon test was used for the statistical analysis (*T11b*-high LUSC $n = 344$, LUAD $n = 1148$, biological replicates), ****$P = 7.96E-78$ (Classical monocyte), ****$P = 4.74E-17$ (Alveolar macrophage), ****$P = 6.04E-133$ (Leukocyte), ***$P = 2.31E-05$ (Non-classical monocyte). The box plots represent the distribution of values for each group, extending from the 25th percentile to the 75th percentile; *T11b*-high LUSC: Classical monocyte (minima=0.00035, median=0.32265, maxima=1, Q1 = 0.16193, Q3 = 0.52806), Alveolar macrophages (minima=0.01914, median=0.44117, maxima=0.91769, Q1 = 0.32846, Q3 = 0.56415), Leukocyte (minima=0.9.87E-05, median=0.41385, maxima=1, Q1 = 0.27138, Q3 = 0.61336), Non-classical monocyte (minima=0.00036, median=0.25014, maxima=0.92007, Q1 = 0.13621, Q3 = 0.44976), LUAD: Classical monocyte (minima=0, median=0.00788, maxima=0.47288, Q1 = 0.00108, Q3 = 0.18980), Alveolar macrophages (minima=0.04437, median=0.36127, maxima=0.67967, Q1 = 0.28261, Q3 = 0.44143), Leukocyte (minima=0, median=0.00264, maxima=0.33064, Q1 = 0.00058, Q3 = 0.13260), Non-classical monocyte (minima=0, median=0.22165, maxima=0.68314, Q1 = 0.10763, Q3 = 0.33783). (C) Gene set enrichment analysis (GSEA) of *Tmprss11b*-high LUSC versus LUAD spatial transcriptomics data with normalized enrichment scores (NES), false discovery rate (FDR) and *P* values for the indicated gene signatures. The nominal *P* and FDR values were obtained from the "GSEA Preranked" tool (from Broad Institute) using a weighted scoring scheme. Gene sets were evaluated based on the default normalized enrichment score method, and statistical significance was determined by bootstrapping with 1000 permutations. (D) Bar graph representing the top M2-like/TAM genes from the differential gene expression (DEG) analysis of *Tmprss11b*-high LUSC versus LUAD spatial transcriptomics data. (E) Representative H&E images and immunohistochemistry (IHC) for SOX2, MSR1 (CD204), HMOX1, and ARG1 in SNL LUSC (top) and mucinous LUAD (bottom). The staining was repeated with lung sections from different mice ($n = 3$–4, biological replicates). Scale bar, 100 μm. Source data are available online for this figure.

Additional studies are needed to fully elucidate the link between TMPRSS11B and macrophage polarization and to determine whether this is solely due to the impact on lactate metabolism or whether additional functions of this enzyme play a role.

Collectively, these findings shed light on an important metabolic regulator of lung squamous cell carcinoma. Given our demonstration that TMPRSS11B promotes tumor growth and an immunosuppressive TME, the development of monoclonal antibodies or small molecule inhibitors that target this enzyme at the surface of LUSCs represents an exciting avenue for the development of new therapeutics for this deadly malignancy.

# Methods

### Reagents and tools table

| Reagent/resource | Reference or source | Identifier or catalog number |
|---|---|---|
| **Experimental models** | | |
| KLN205 cells | ATCC | CRL-1453 |
| Rosa26^LSL-SOX2-IRES-GFP; Nkx2-1^fl/fl;Lkb1^fl/fl (SNL) | Mollaoglu et al, 2018 | |
| **Recombinant DNA** | | |
| pLentiCRISPR V2 GFP | Addgene | 82416 |
| pLentiCRISPR V2 Puro | Addgene | 98290 |
| pLentiCRISPR V2 Cre | Addgene | 82415 |
| psPAX2 | Addgene | 12260 |
| pMD2.G | Addgene | 12259 |
| **Antibodies** | | |
| Phospho-p44/42 MAPK (Erk1/2) (Thr202/Tyr204) | Cell Signaling | 4370S |
| CD4 | Cell Signaling | 25229S |
| SOX2 | Cell Signaling | 23064S |
| CD204 (MSR1) | Invitrogen | MA5-29733 |
| HMOX1 | Abcam | ab189491 |

| Reagent/resource | Reference or source | Identifier or catalog number |
|---|---|---|
| CD8α | Cell Signaling | 98941S |
| ARG1 | Cell Signaling | 93668S |
| KRT16 | Proteintech | 17265-1-AP |
| LYPD3 | Invitrogen | PA5-48085 |
| LY6G | InVivoMab | BE0075-1 |
| MPO | Abcam | ab208670 |
| CD11b | Abcam | ab133357 |
| MCT4 | Abcam | ab244385 |
| GLUT1 | Cell Signaling | 73015S |
| CD31(PECAM-1) | Cell Signaling | 77699S |
| MCT1 | Abcam | ab85021 |
| **Oligonucleotides and other sequence-based reagents** | | |
| **Genotyping primers for SNL GEMM model** | **Target** | **Sequence** |
| WS268 (Forward) | Sox2 | GTT ATC AGT AAG GGA GCT GCA GTG G |
| WS270 (Reverse-Flox) | Sox2 | AAG ACC GCG AAG AGT TTG TCC TC |
| WS271 (Reverse-wt) | Sox2 | GGC GGA TCA CAA GCA ATA ATA ACC |
| NKX (Forward) | Nkx2-1 | TTT CTC TCT TGC GGG CTC TA |
| NKX (Reverse) | Nkx2-1 | GGA GGG GCG AGT AGA GAG AG |
| 11FWD (Forward) | Lkb1 | ATC GGA ATG TGA TCC AGC TT |
| 11REV (Reverse) | Lkb1 | ACG TAG GCT GTG CAA CCT CT |
| **Primers for surveyor assay** | | **Sequence** |
| mT11B_Sur_1_FP | | AGAGACTCTTGGGGATGCTG |
| mT11B_Sur_1_RP | | GTGTGGGATAGGATGGAGGA |
| mT11B_Sur_2_FP | | ACTGCTCTTCCAGGGTTCCT |
| mT11B_Sur_2_RP | | ACCAGGACTTGTGAGGATGG |
| mT11B_Sur_3_FP | | ACTTGCCCTCTTCTGGTGTG |
| mT11B_Sur_3_RP | | CAACTGCAGGTGGTTTAGAGTC |

| Reagent/resource | Reference or source | Identifier or catalog number |
|---|---|---|
| mT11B_Sur_4_FP | | TATGGGGTTGGAGAAATGGA |
| mT11B_Sur_4_RP | | AGAGCTGCCTTCCACACAGT |
| mT11B_Sur_5_FP | | TGCAATGGGCTATAATGCAA |
| mT11B_Sur_5_RP | | CTTGCCCCCAAACTAATGAA |
| **Guide RNA (sgRNA)** | | **Sequence** |
| Tmprss11b sg1 | | GAATGGTAAACACTACTGTG |
| Tmprss11b sg2 | | TTGAACAGAATGCTGCATGT |
| Tmprss11b sg3 | | ACCTACGACAGAATCACGGG |
| **shRNA** | | **Sequence** |
| Tmprss11b sh1 | | CAAGGTACCAAATATCTCG |
| Tmprss11b sh2 | | TTCCGGAAGACAAACCCGA |
| Tmprss11b sh3 | | TAATCTTCTCAGCATCAAC |
| **Sequencing Primers** | | **Sequence** |
| CMV-Forward | | CGCAAATGGGCGGTAGGCGTG |
| hU6-Forward | | GAGGGCCTATTTCCCATGATT |
| WPRE-Reverse | | CATAGCGTAAAAGGAGCAACA |
| T7-Forward | | TAATACGACTCACTATAGGG |
| **Chemicals, enzymes, and other reagents** | **Reference or source** | **Identifier or catalog number** |
| 2,2,2-Tribromoethanol | Sigma-Aldrich | T48402-25G |
| 2-Methyl-2-butanol | Sigma-Aldrich | 152463 |
| Puromycin | Gibco | A1113803 |
| Doxycycline | RPI | D43020-250.0 |
| Doxycycline hyclate | Millipore Sigma | D9891-10G |
| RNAseOUT | Millipore Sigma | R2020-250ML |
| Tween 20 | Millipore Sigma | P9416-100ML |
| Hexadimethrine bromide | Millipore Sigma | 107689 |
| Fetal bovine serum (FBS) | Millipore Sigma | F2442 |
| Effectene Transfection Reagent | Qiagen | 301427 |
| RNeasy Mini Kit | Qiagen | 74106 |
| Opal 3-Plex Detection kit | Akoya Biosciences | NEL810001KT |
| RNAscope Multiplex Fluorescent Kit v2 | Advanced Cell Diagnostics | 3231100 |
| Guide-it Mutation Detection kit | Takara | 631448 |
| TaqMan Universal qPCR Master Mix | ThermoFisher | 4304437 |
| **Software** | **Source** | **Website** |
| Prism 10 | GraphPad | https://www.graphpad.com/features |

| Reagent/resource | Reference or source | Identifier or catalog number |
|---|---|---|
| Fiji (ImageJ) | NIH | https://imagej.net/ |
| Zen Blue | Zeiss | https://www.zeiss.com/microscopy/us/products/software/zeiss-zen.html |
| GSEA | Broad Institute | https://www.gsea-msigdb.org/gsea/index.jsp |
| Phenochart | Akoya Biosciences | https://www.akoyabio.com/support/software/ |
| Loupe Browser | 10X Genomics | https://www.10xgenomics.com/support/software/loupe-browser/latest |
| **Other** | | |

## Mice

*Rosa26-Sox2-Ires-Gfp<sup>LSL/LSL</sup>;Nkx2-1<sup>fl/fl</sup>;Lkb1<sup>fl/fl</sup>* $Rosa26\text{-}Sox2\text{-}Ires\text{-}Gfp^{LSL/LSL};Nkx2\text{-}1^{fl/fl};Lkb1^{fl/fl}$ (SNL) mice were provided by Dr. Trudy Oliver (Duke University). The mice were maintained on a mixed background through intercrosses. DBA/2 mice were purchased from The Charles River Laboratory.

## Ethics statement

Mice were monitored closely throughout all experimental protocols to minimize discomfort, distress, or pain. If any signs of pain and distress were detected (disheveled fur, decreased feeding, significant weight loss (> 20% body mass), limited movement, or abnormal gait), the animal was removed from the study and euthanized. All procedures involving mice were performed in accordance with the recommendations of the Panel on Euthanasia of the American Veterinary Medical Association and protocols approved by the UTSW Institutional Animal Care and Use Committee.

## Cell culture

KLN205 murine lung squamous cell carcinoma cells (from Dr. Rolf Brekken) were cultured in Gibco™ DMEM media (Cat: 11995065) supplemented with 10% FBS and 1% antibiotic and antimycotic (Cat: 15240062). The human basal cell line BEAS-2B was acquired from the Duke Cell Culture Facility (Sigma Catalog #95102433). Cells were cultured in 2D adherent cell plates in Bronchial Epithelial Cell Growth Basal Medium (Lonza Catalog # CC-3171) with Supplements and Growth Factors (Lonza Catalog # CC-4175) along with 0.2% Penstrep (BEBM media). For passaging, 0.25% Trypsin-EDTA (Gibco catalog #25200056) was used to dislodge cells. HEPES Buffered Saline Solution (Catalog # CC-5024) with 15% FBS was used to stop the reaction. Cells were centrifuged at 250 g for 5 min at 4 °C. Cells were resuspended in BEBM media and plated. All cell lines have been DNA fingerprinted using the PowerPlex Fusion 24 kit (Promega) and have been found to be *mycoplasma* free using a direct PCR method with GoTaq Green Master Mix (Promega, M712).

## Plasmids

LentiCRISPR version 2, PAX2, and MD2 plasmids were obtained from Addgene (52961, 12260, and 12259). sgRNA sequences targeting *Tmprss11b* were selected from the Brie sgRNA library

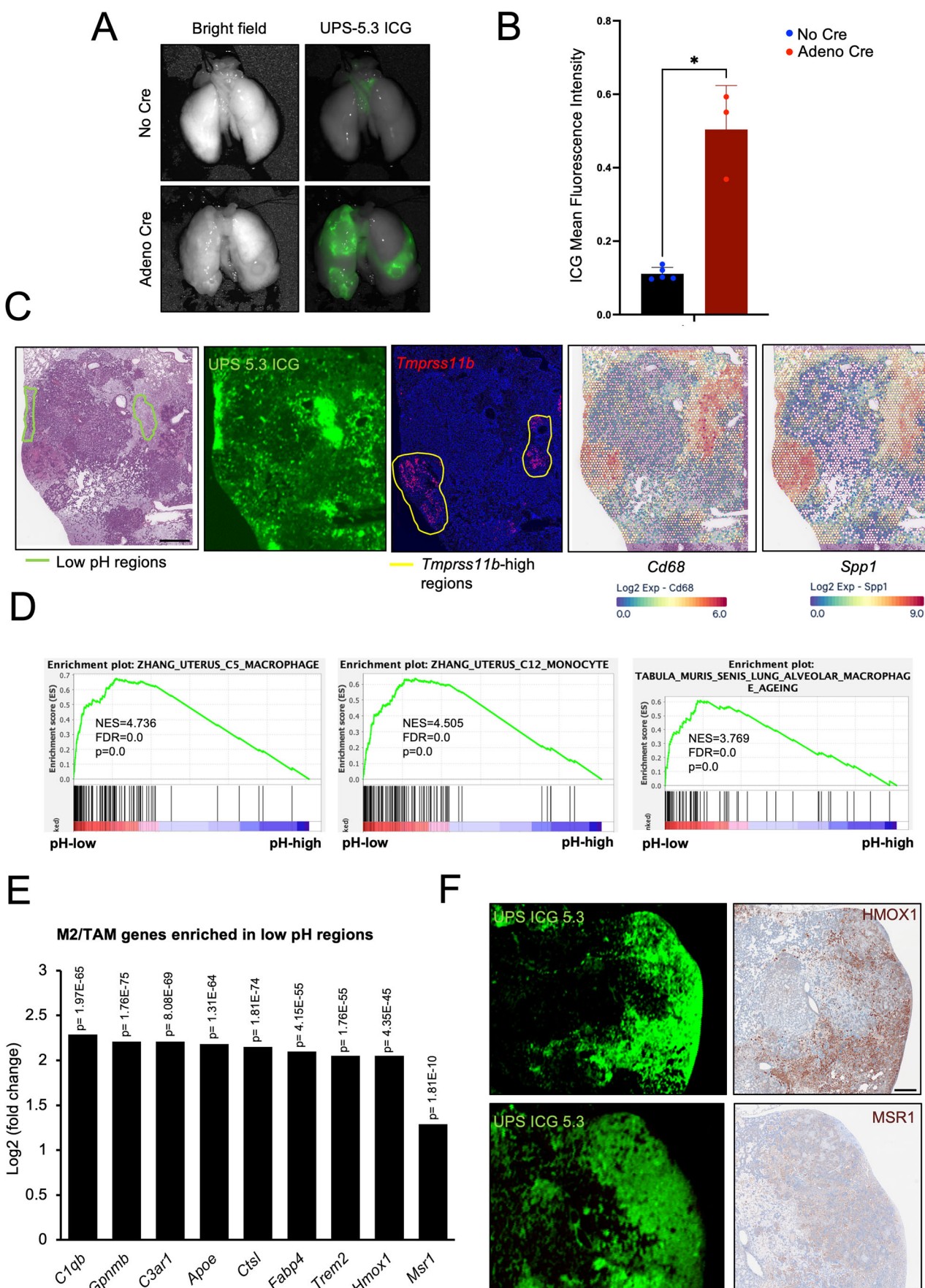

**Figure 6. *Tmprss11b*-high squamous tumors and acidified regions of the TME are enriched for immunosuppressive macrophages.**

(A) Representative ex vivo images of lungs from SNL mice, control (no Cre) or infected with Ad-Cre (11 months post infection), 24 h post injection with PDBA ICG 5.3 ultra pH-sensitive (UPS) nanoparticle. Bright-field images (left) and the corresponding ICG fluorescence images (right) are shown. (B) Quantification of mean ICG fluorescence intensity (control $n = 5$ mice; Ad-Cre $n = 3$ mice, biological replicates). Unpaired $t$ test with Welch's correction was used for the analysis, $P = 0.0281$. Plot represents mean ± SD. (C) H&E image of the lung section from Fig. 3A), with ICG signal on the same section, and RNAscope of *Tmprss11b* (red) on a serial section. Also shown are spatial plots depicting enrichment of *Cd68* (macrophage marker) and *Spp1* (immunosuppressive macrophage marker) transcripts in these regions. Scale bar, 1 mm. (D) Gene set enrichment analysis (GSEA) of immune cell pathways from the low pH versus high pH spatial transcriptomics data with normalized enrichment scores (NES), false discovery rate (FDR) and $p$ values for the indicated gene signatures. The nominal $P$ and FDR values were obtained from the "GSEA Preranked" tool (from Broad Institute) using a weighted scoring scheme. Gene sets were evaluated based on the default normalized enrichment score method, and statistical significance was determined by bootstrapping with 1000 permutations. (E) Bar graph representing top M2-like/TAM genes from the differential gene expression (DEG) analysis of low pH versus high pH spatial transcriptomics data. Differential gene expression was calculated using Seurat's FindAllMarkers function, and direct comparisons between two classes and corresponding $P$ values were obtained using Seurat's FindMarkers function with the Wilcoxon rank-sum test with Bonferroni $P$ value correction. (F) Representative ICG fluorescence images (left) and the corresponding immunohistochemistry (IHC) validation of HMOX1 and MSR1 (right) in SNL lung sections. The staining was repeated with lung sections from different mice ($n = 2$, biological replicates). Scale bar, 500 µm. Source data are available online for this figure.

(target gene ID: 319875). The SMARTvector inducible lentiviral shRNA plasmids were obtained from Dharmacon. The sgRNAs and shRNAs sequences are provided in the "Reagents and Tools table".

## Generation of *Tmprss11b* knockout cells

HEK293T ($1 \times 10^8$) cells were co-transfected with lentiCRIPSR version 2 (1 µg), PAX2 (666 ng) and MD2 (333 ng) helper plasmids using Effectene Transfection Reagent (Qiagen, Cat: 301425). Lentiviral supernatant was collected 48 h post-transfection and filtered. Recipient KLN205 cells were infected with viral supernatant containing 8 µg/ml polybrene (Sigma-Aldrich) and replenished with fresh media. After 48 h, the transduced cells were cultured in fresh media containing 2 µg/ml puromycin for 7–9 days.

## Generation of inducible *Tmprss11b* knockdown cells

HEK293T ($1 \times 10^8$) cells were co-transfected with SMARTvector inducible lentiviral shRNA plasmid (1 µg), PAX2 (666 ng), and MD2 (333 ng) helper plasmids using Effectene Transfection Reagent (Qiagen, Cat: 301425). Lentiviral supernatant was collected 48 h post-transfection and filtered. Recipient KLN205 cells were infected with viral supernatant containing 8 µg/ml polybrene (Sigma-Aldrich) and replenished with fresh media. After 48 h, transduced cells were selected in fresh media containing 2 µg/ml puromycin for 7–9 days and cultured in 2–3 µg/ml doxycycline. shRNA expression was monitored over 4 days using the turbo RFP expression. At 4 days post-induction, the cells were harvested for RNA to assess the knockdown.

## KLF4 overexpression

Human *KLF4* was overexpressed in BEAS-2B immortalized human lung epithelial cells under the control of a tetracycline response element (TRE). *KLF4* overexpression was induced in cells using 250 ng/mL of doxycycline. KLF4 sequence was validated through Plasmidsaurus. Total RNA was extracted after 48 h of doxycycline treatment, and a TRE-GFP expression construct was used as a control. RNA was extracted using a RNeasy mini kit (Qiagen Cat. No. 74104).

## Tumorigenesis assays

KLN205 cells ($3 \times 10^5$) expressing non-targeting sgRNA or *Tmprss11b* sgRNA were injected subcutaneously into the right flanks of 6- to 8-week-old DBA/2 female mice (Charles River). Tumor volumes were measured every 3 days using calipers until the average tumor mass reached 600 mm$^3$. sgRNA sequences are provided in "Reagents and Tools table".

For inducible knockdown, KLN205 cells ($5 \times 10^5$) expressing scrambled shRNA or *Tmprss11b* shRNA were injected subcutaneously into the right flanks of 6- to 8-week-old DNA/2 female mice (Charles River). Tumor volumes were measured every 3 days using calipers. When the average tumor volume across all the study groups reached 100 mm$^3$, mice were maintained on doxycycline water (2 g/L doxycycline and 2% sucrose) for the duration of the experiment.

Tumor volumes were calculated using the formula (length × width$^2$)/2. At the terminal time point, the tumors were resected for downstream analysis.

## RNA extraction and quantitative real-time PCR (qRT-PCR) analysis

Total RNA was isolated from cells using RNeasy Mini Kit (Qiagen). Total RNA was isolated from tumor tissues using Trizol (Ambion by Life Technologies; 15596018), followed by additional cleanup and DNase digestion using RNeasy Mini Kit (Qiagen). For qRT-PCR of mRNA, complementary DNA (cDNA) synthesis was performed with 1–5 µg of the total RNA for reverse transcription using Superscript IV VILO Master Mix (5x) (Invitrogen, Cat: 11756050). mRNA expression was assessed using TaqMan probes (Invitrogen) corresponding to *Tmprss11b* (Mm00621706_m1, Mm00621702_m1, and Mm00621704_m1) and *Gapdh* (Mm99999915_g1) and calculated using the $2^{ddCt}$ method.

## RNA-sequencing and analysis

RNA was isolated from tumor tissues as indicated in the last section. The sequencing was performed at the McDermott Center Next Generation Sequencing Core at UT Southwestern Medical Center, and the analysis was performed at the McDermott Center Bioinformatics lab. Fastq files were quality checked using fastqc (v0.11.2)(Andrews, 2010). Read from each sample were mapped to the reference genome using STAR (v2.5.3a) (Dobin et al, 2013). Read counts were generated using featureCounts (Liao et al, 2014) and the differential expression analysis was performed using edgeR (Robinson et al, 2010).

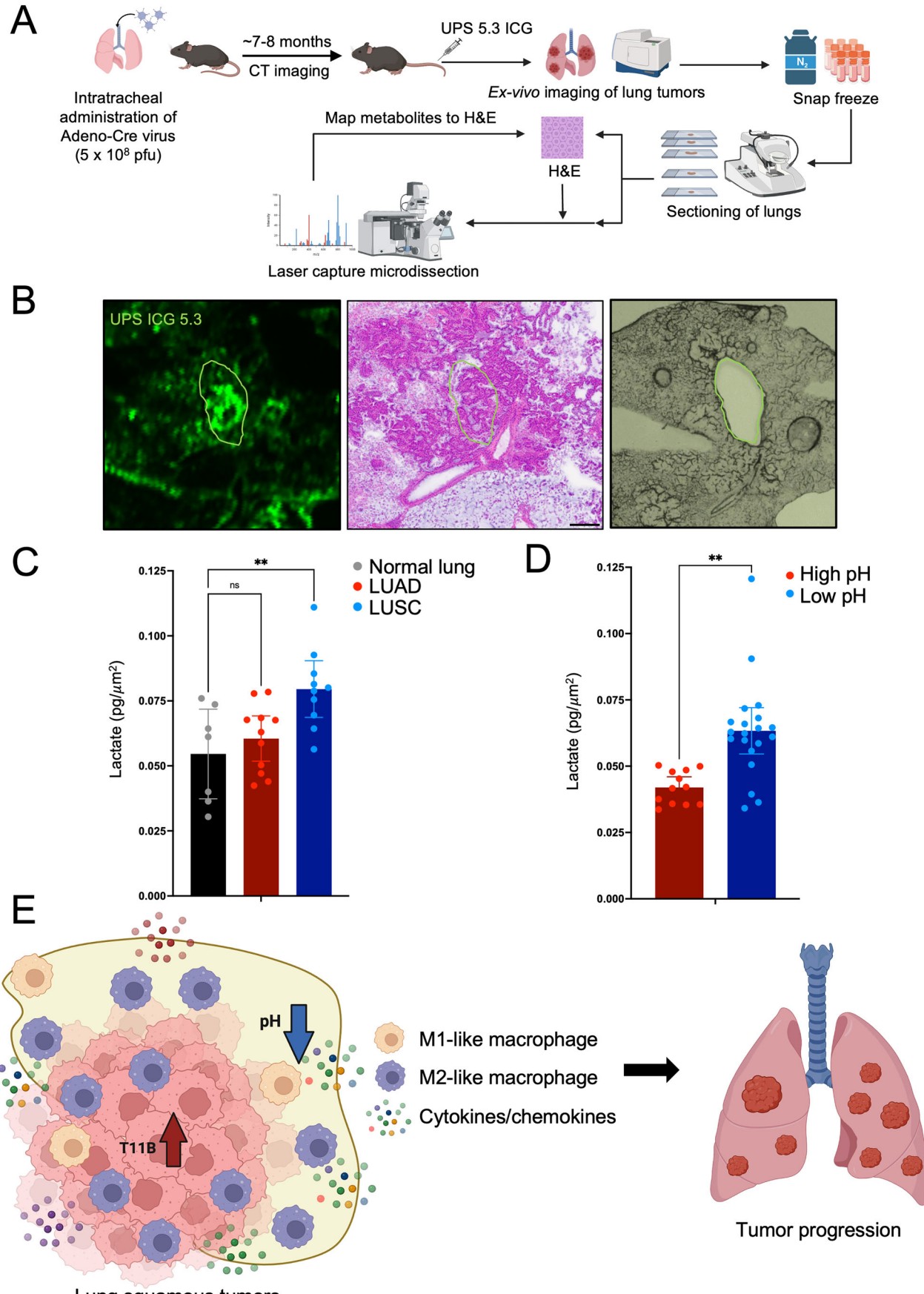

◄ **Figure 7. *Tmprss11b*-high squamous tumors and acidified regions have elevated levels of lactate.**

(A) Schematic representation of the experimental pipeline to assess metabolites in SNL lung tumors through laser capture microdissection (LMD) followed by mass spectrometry. Figure created in BioRender. (B) Representative ICG fluorescence images (depicting nanoparticle accumulation) (green), H&E image (middle), and bright-field image of the serial section (right) used for LMD with the same region annotated in green. Scale bar, 200 μm. (C) Quantification of lactate concentration (pg/μm²) in SNL lung tissues; normal, LUADs and LUSCs ($n = 7$–11 regions per group from a total of 2 mice, biological replicates). Ordinary one-way ANOVA with Dunnett's multiple comparisons test was used for the statistical analysis, $P = 0.0057$. The plot represents mean with 95% CI. (D) Quantification of lactate concentration (pg/μm²) in the regions of low acidity (high pH) and high acidity (low pH), in the SNL lung tissues ($n = 12$–20 regions per group from a total of two mice, biological replicates). Ordinary one-way ANOVA with Dunnett's multiple comparisons test was used for the statistical analysis, $P = 0.0013$. The plot represents the mean with 95% CI. (E) Schematic representation of the immunosuppressive niche established in *Tmprss11b*-high lung squamous tumors and the surrounding acidified regions in the tumor microenvironment. Figure created in BioRender. Source data are available online for this figure.

## Intratracheal administration of adenovirus and tissue collection

Cre-expressing adenovirus (Ad-Cre) was purchased from Viral Vector Core Facility (University of Iowa). *Rosa26-Sox2-Ires-Gfp^LSL/LSL^;Nkx2-1^fl/fl^;Lkb1^fl/fl^* (SNL) mice (males and females) 6- to 8-weeks of age were intratracheally administered Ad-Cre at $1$–$10 \times 10^8$ pfu/mouse.

Mice were euthanized by intraperitoneal administration of an overdose of Avertin at the indicated time points. Lungs were inflated and perfused through the trachea with 4% paraformaldehyde (PFA), fixed overnight, washed in 1× PBS for 24–48 h, then transferred to 70% ethanol and subsequently embedded in paraffin. Sections were cut and stained with H&E by the UTSW HistoPathology Core.

## Immunohistochemistry

The formalin-fixed paraffin-embedded (FFPE) slides were de-paraffinized with xylenes and hydrated with ethanol washes. Slides were then treated with either Citrate-based (pH 6.0) or Tris-based (pH 9.0) Antigen Unmasking Solution (Vector Lab, H-3300 or H3301). Antigen retrieval was performed using a steamer, followed by blocking using BLOXALL™ Endogenous Blocking Solution (Vector Lab, SP-6000) and washed in 1× PBS. Then the slides were incubated with 2.5% Normal Goat Serum (Vector Lab, S1012) followed by incubation with appropriate primary antibodies diluted in 2% bovine serum albumin (BSA) (in PBS) at 4 °C overnight. After extensive washing with either PBS/TBS, the slides were incubated with ImmPRESS ® HRP Goat Anti-Rabbit IgG (Vector Lab, MP7451) or Goat Anti-Rat IgG (Vector Lab, MP7404) at room temperature (RT) for 30 min. The signal was developed with ImmPACT ® DAB Substrate (Vector Lab, SK-4105), and sections were counterstained with hematoxylin (Vector Lab, H-3404), and mounted using Prolong™ Diamond Antifade Mountant (Invitrogen, Cat: P36965). Whole slide images were captured using Hamamatsu NanoZoomer S60 at the UTSW Whole Brain Microscopy Facility. Antibody information is provided in the "Reagents and Tools table".

## Multiplex fluorescent immunohistochemistry and quantification

The fluorescent immunohistochemistry was performed using Akoya Opal 3-Plex or 6-plex Detection kit (NEL810001KT, NEL811001KT) following the manufacturer's instructions. Briefly, FFPE slides were de-paraffinized using xylenes, hydrated with ethanol washes and then fixed with 4% paraformaldehyde (PFA).

Antigen retrieval was performed at either pH 6.0 or pH 9.0 using a microwave. Blocking was performed using Antibody Diluent/Block from the kit. Slides were then incubated with the primary antibodies at 4 °C overnight. After extensive washing with TBST, the slides were incubated with Opal Anti-Ms+Rb HRP and then the appropriate Opal/TSA plus fluorophore. For multiplexing, this was followed by another round of antigen retrieval, and same steps were followed as before until all the targets have been achieved. The slides were then incubated with Spectral DAPI, followed by mounting with Prolong™ Diamond Antifade Mountant (Invitrogen, Cat: P36965). Whole slide images were captured using Zeiss Axioscan.Z1 at the UTSW Whole Brain Microscopy Facility and Vectra Polaris™ system from Akoya Biosciences.

For quantification, the Zen Blue software from Zeiss was used. For each slide, 10–14 equally sized fields are selected across the slide and mean fluorescent intensities (MFI) for each channel was calculated for each field.

## RNA in situ hybridization

RNA in situ hybridization was performed on formalin-fixed paraffin-embedded (FFPE) tissue sections following instructions from the RNAscope Multiplex Fluorescent Kit v2 (Cat. No. 3231100, Advanced Cell Diagnostics) using the following probes purchased from Advanced Cell Diagnostics: *Tmprss11b* (Cat No. 1268741-C2), *Sox2* (Cat No. 401041-C3), and negative control *DapB* (Cat No. 310043-C2). Slides were counterstained with DAPI (Cat No. 320858, Advanced Cell Diagnostics) and mounted with Prolong™ Diamond Antifade Mountant (Invitrogen, Cat: P36965). Whole slide images were obtained using Axioscan.Z1 and Vectra Polaris™ imagers.

## Ultra pH-sensitive (UPS) nanoparticle imaging

Ultra pH-sensitive (UPS) nanoparticles were used for imaging acidic regions in mouse lung tumors and other tissues (Feng et al, 2019). Briefly, SNL mice bearing lung tumors (from CT imaging) were injected with 2.5 mg/kg of UPS nanoparticle PDBA-ICG 5.3. 24 h post-injection, the mice were euthanized by intraperitoneal administration of an overdose of Avertin. Lungs were inflated and perfused through the trachea with 4% paraformaldehyde (PFA). The lungs were then imaged with liver, spleen, kidney, heart, and brain from the same mouse for brightfield (BF) and Indocyanine green (ICG) signals using Pearl ® Trilogy small animal imaging system (LICOR Bio). Post imaging, the lungs were fixed overnight in 4% PFA, washed in 1× PBS for 24–48 h, then transferred to 70% ethanol and subsequently embedded in paraffin. Sections were cut

and stained with H&E. The ICG signal in lungs was quantified as mean fluorescent intensity (MFI) using the Image Studio™ software (LICOR Bio). For the analysis, the ICG signal in the brain was used as background and normalized for lungs and other tissues accordingly.

## MRI and CT imaging

Magnetic resonance imaging (MRI) was performed on mice using the 7 T Bruker Biospec instrument at the Advanced Imaging Research Center at UT Southwestern with the help of Dr. Janaka Wansapura.

Computed tomography (CT) imaging was performed on mice using the X-Cube CT instrument from Molecubes at the Pre-clinical Radiation Core Facility at UT Southwestern.

## Surveyor assay

Surveyor assay was used to assess the efficiency of the different *Tmprss11b* targeting sgRNAs. The assay was performed on *Tmprss11b* knockout and non-targeting control KLN205 cells using Guide-it™ Mutation Detection kit (Takara, Cat No. 631448) following the manufacturer's protocol. Briefly, genomic DNA was isolated from the above-mentioned cells, and the mutated regions were amplified using specifically designed surveyor primers (provided in "Reagents and Tools table"). Then the amplified sequences were subjected to cleavage by the Guide-it resolvase and screened by running samples on agarose gel electrophoresis.

## Spatial transcriptomics: Visium CytAssist

Hematoxylin and eosin-stained tissue slides were scanned at ×40 magnification using the Leica Biosystems Aperio AT2 DX bright-field slide scanner. Regions of lung adenocarcinoma and lung squamous cell carcinoma morphology were annotated. The regions of interest for spatial transcriptomics were digitally annotated using QuPath (v.5.0.1) and exported as a TIFF image file.

Tissue slides were decoverslipped, washed, and rehydrated following the 10x Genomics Visium CyAssist Tissue Preparation Guide (CG000518 rev. B). Slides were immersed in xylene, then placed on a metal block cooled with dry and the coverslip was removed with a razor blade. Sections were washed and rehydrated by sequential immersions in xylene, 100% ethanol, 96% ethanol, and 70% ethanol, according to manufacturer recommendations. A region of the tissue outside of the region of interest was removed for RNA quality control. RNA was isolated with the Qiagen RNeasy Mini Kit (cat: 74104), and quality was evaluated using the Agilent Technologies RNA 6000 Pico kit (cat: 5067-1513) with a DV200 assay in the Agilent 2100 BioAnalyzer.

Rehydrated samples were destained and decrosslinked following 10x Genomics protocol CG000520 rev B. Sample preparation was performed with a 10x Genomics Visium CytAssist Spatial Gene Expression for FFPE, Mouse Transcriptome, 6.5 mm kit (PN-1000521) and its corresponding protocol (GC000495 rev. C). Mouse whole-transcriptome left and right-hand probes were hybridized to the tissue at 50 °C for 18 h. Hybridized probes were ligated at 37 °C for 60 min. Tissue was then stained with 10% eosin Y for one minute and rinsed with 1× PBS. Slides were placed in the Visium CytAssist instrument and manually aligned for capture of the regions of interest. Gene expression probes were released from tissue and captured by the Visium CytAssist spatial slide using the CytAssist instrument at 37 °C for 30 min. The spatial slide was washed with 2× SSC, and captured probes were extended on the slide at 45 °C for 15 min. Probe amplicons were eluted from the capture slide with 0.08 M KOH and neutralized with Tris-HCl pH 8.0. Sample probes were pre-amplified by PCR, and optimal cycle determination was performed using qPCR. Final sequencing libraries were prepared using 10x Genomics dual index plate TS set A (PN-3000511) and sequenced with read depth of at least 50,000 reads per spot using the Singular Genomics G4 platform with an F3 flow cell.

## Spatial transcriptomics analysis pipeline

Visium spatial transcriptomics reads were processed using 10x Genomics Space Ranger v2.0.1. Samples were post-processed using R v.4.3.1 (R Core Team, 2020) with Seurat v. 5.0.3 (Hao et al, 2024). Spots corresponding to Adenocarcinoma, Squamous cell carcinoma, and low pH were annotated using Loupe Browser v.6.5.0 based on morphology from H&E images and corresponding assays in adjacent tissue sections. Samples were processed for quality control, preserving spots with a minimum of 700 read counts and 200 genes, and genes present in at least five spots. RNA gene expression libraries were normalized using SCtransform (Hafemeister and Satija, 2019).

Spots expressing *Tmprss11b* at or above the 95th percentile were assigned "Tmprss11b-High". Squamous tissue regions were further subclassified according to dichotomized *Tmprss11b* expression. All spots were categorized into "*Tmprss11b*-High Squamous", "*Tmprss11b*-Low Squamous", "Adenocarcinoma", and "Other" classes based on the histology. Furthermore, the spots were grouped into "low pH" regions, defined as the regions with the highest accumulation of the ICG nanoparticle, and "high pH" regions, defined as the remaining area excluding the "low pH" regions. Differential gene expression among all these annotated regions was calculated using Seurat's FindAllMarkers function, and direct comparisons between two classes were performed with Seurat's FindMarkers function. Differentially expressed genes were used to perform pathway enrichment analysis using clusterProfiler v.4.10.0 (Wu et al, 2021; Yu et al, 2012) using Gene Ontology (Ashburner et al, 2000; Central et al, 2023), KEGG references (Kanehisa et al, 2023; Kanehisa and Goto, 2000). Gene expression signatures for "Mouse_Hillock", "Mouse_Basal", "Mouse_Mucinous", "Glycolysis", and "TCA cycle" (Dataset EV1) phenotypes were evaluated for each spot using Seurat's AddModuleScore function.

## Tangram cell type deconvolution

Spatial transcriptomics technologies provide spatially resolved gene expression data; however, their resolution is often lower than that of single-cell RNA sequencing (scRNA-seq). Consequently, each spatial spot or voxel may contain contributions from multiple cell types, making it challenging to determine the precise cellular composition of a tissue. Spatial deconvolution addresses this issue by integrating high-resolution scRNA-seq data with lower-resolution spatial transcriptomics data to infer the relative proportions of different cell types at each spatial location. Among

the available spatial deconvolution methods (Cable et al, 2022; Dong and Yuan, 2021; Elosua-Bayes et al, 2021; Kleshchevnikov et al, 2022), we employed Tangram (Cable et al, 2022) due to its ability to map single cells onto spatial transcriptomics data across multiple platforms with high accuracy. Tangram utilizes deep learning-based optimal transport to achieve unbiased alignment, multimodal integration, and gene-level precision. By leveraging both scRNA-seq and spatial transcriptomics datasets, Tangram optimally assigns single cells to spatial locations while preserving gene expression consistency across modalities.

### Data preprocessing

We obtained reference single-cell RNA sequencing (scRNA-seq) data from The Tabla Muris Consortium (*Nature* 2018) (Schaum et al, 2018) and spatial transcriptomics (ST) data from relevant datasets. The scRNA-seq dataset was preprocessed by normalizing gene expression counts and performing dimensionality reduction using principal component analysis (PCA). Cells were then clustered based on their transcriptional profiles, and each cluster was annotated with its corresponding cell type.

### Mapping scRNA-seq to spatial transcriptomics

Tangram applies a probabilistic framework to align single-cell gene expression profiles with spatial transcriptomics data by optimizing the mapping based on a reconstruction loss function. The spatial mapping was performed using the following steps:

1. Model Initialization: The single-cell and spatial gene expression matrices, along with cell-type annotations, were input into Tangram, ensuring that only genes common to both datasets were retained.
2. Optimization: Tangram estimated the optimal mapping by solving a constrained optimization problem that minimizes the reconstruction error while preserving gene expression consistency between single-cell and spatial transcriptomics data.
3. Cell-Type Decomposition: Using the trained model, Tangram inferred the proportion of each cell type at individual spatial locations, generating a probabilistic estimate of cellular distributions across the tissue.

The top 20 differentially expressed genes used for alignment with the Visium data, effectively serving as marker genes for each cell type are provided in Table EV1.

### Laser capture microdissection

The SNL mice administered with Ad-Cre were euthanized at 8-months post infection by intraperitoneal administration of an overdose of Avertin. The lungs were flash-frozen and sectioned into ~50-µm-thick sections on polyethylene terephthalate membrane (PTFE) slides. Leica laser LDM6 microdissection microscope was used to selectively cut the tissue areas exclusively identified as lung squamous cell carcinoma, lung adenocarcinoma, adeno-squamous, low pH, high pH, and normal tissue regions based on adjacent histology staining and fluorescent UPS probe distribution. Each excised area was collected in individual PCR vials. 100 µL of 0.1% formic acid in water was added to each PCR vial to extract water-soluble metabolites. The vials were then subjected to sonication for 30 min in a water bath followed by 20 min vortex at 2500 rpm

(VWR DVX-2500 multi-tube vortex mixer). The samples were centrifuged at 14,000×g for 10 min. Then 45 µL of the supernatant was transferred into an autosampler vial to mix with 5 µL internal standard (IS) mix (5 µg/mL). The sample injection volume was 10 µL.

A Shimadzu CBM-20A Nexera X2 series LC system (Shimadzu Corporation, Kyoto, Japan) equipped with degasser (DGU-20A) and binary pump (LC-30AD) along with autosampler (SIL-30AC) and (CTO-30A) column oven was used. Chromatographic separation of glucose, lactic acid, pyruvic acid, glutamic acid, succinic acid and citric acid were achieved using a Phenomenex Luna C8 (2), 5 µm 100 Å, 150 × 2.0 mm column. 0.1% formic acid in water was used for mobile phase A, and 0.1% formic acid in acetonitrile was used as mobile phase B. The LC flow rate was set at 0.3 mL/min with gradient started from 3% of B for 1 min, 30% B by 5 min, 98% B by 5.5 min maintained to 7 min, then switched back to 3% B by 7.1 min and maintained to 8 min. The autosampler was maintained at 5 °C. Injection volume of 10 µL was used.

The primary stock solutions for all analytes including standards and isotope labeled internal standards (IS) (1.0 mg/mL) were prepared in LC/MS grade water and subsequent dilutions were prepared in 0.1% formic acid in water. Standard curves were prepared at a range of concentrations at 1, 5, 10, 25, 50, 100, 500, 1000, 5000, and 10,000 ng/mL with different lowest calibration concentration points suited for different metabolites. IS mix of 5 µg/mL of each D-glucose (U-$^{13}C_6$), L-Lactic acid-$^{13}C_3$, Pyruvic acid sodium salt $^{13}C_1$, L-Glutamic acid-$^{13}C_5$, Succinic acid -2,2,3,3-D4, and Citric acid 2,2,4,4-D4 was also prepared in 0.1% formic acid in water. 5 µL of IS mix (5 µg/mL) was spiked in 45 µL of standard curves and samples to correct for any response-based differences created from the instrument or sample preparation.

An AB Sciex (Foster City, CA, USA) 6500 + QTRAP mass spectrometer, equipped with a Turbo ion spray™ (ionization source) was used as the detector. The mass spec interface temperature was set at 500 °C. The ion spray voltage was set at −4500 Volts. Other parameters such as nebulizer gas, curtain gas, auxiliary gas, and CAD gas were set at 50, 55, 65, and Medium, respectively. Detection of the ions was performed in multiple reaction monitoring (MRM) mode, the transition pairs Q1/Q3 were set on unit resolution. MRM transition pairs and each of their entrance potential (EP), collision energy (CE), and collision exit potential (CXP) are presented in Table EV2.

The LC/MS/MS data were processed by Analyst software (version 1.7.3). The results were fitted to linear regression analysis using $1/X^2$ as a weighting factor. Quantifier ions Q3 are listed in Table EV2. The final concentration of metabolites was calculated by normalizing the measured amounts to the area of the dissected tissue regions.

### Functional analysis of gene sets

Pathway and network analysis were performed using GSEA 4.3.2 application from the Broad Institute (Mootha et al, 2003; Subramanian et al, 2005). The GSEA Preranked tool was used for ranked gene list using the rank_score=sign_of_FC*−log(P val) for all the expressed genes in the RNA-seq dataset with a weighted scoring scheme and using rank_score=log2FC for all the genes showing significant differences in expression (P val <0.05) in the spatial transcriptomics dataset with a weighted scoring scheme. The

nominal *P* and FDR values were obtained from the "GSEA Preranked" algorithm using number of permutations = 1000 and normalization method = meandiv.

## Statistics and reproducibility

An unpaired *t* test with Welch's correction was used for comparisons between two groups (for comparing MFI from IHC-F for CD4 + T cells and other indicated analyses). Ordinary one-way ANOVA with Dunnett's multiple comparisons test or Brown–Forsythe and Welch ANOVA test with Dunnett's T3 multiple comparisons test was used for comparisons between more than two groups (for qRT-PCR analyses, comparing tumor volumes from implantation studies and other indicated analyses). For the tumorigenesis assays in syngeneic mice, linear mixed-effects models were used to investigate if there were significant differences in tumor volume over time among the three groups. A Wilcoxon rank-sum test was used to compare the distribution of *Tmprss11b* expression across tissue regions.

## Data availability

The datasets produced in this study are available in the following databases: RNA sequencing data: Gene Expression Omnibus (GEO) GSE292085. Spatial transcriptomics data: Gene Expression Omnibus (GEO) GSE292706. Imaging data: BioImage Archive (BioStudies) S-BIAD2301.

The source data of this paper are collected in the following database record: biostudies:S-SCDT-10_1038-S44319-025-00631-1.

## Peer review information

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

## Acknowledgements

We thank Joshua Mendell and members of the O'Donnell laboratory for critical reading of the manuscript. KAO is supported by the NCI (R01CA273585, R01CA207763, and P50CA70907), the Cancer Prevention Research Institute of Texas (CPRIT RP190610, RP200327, RP250391, and RP250572), the Welch Foundation (I-1881), the V Foundation (T2021-011), the Department of Defense (DoD LC190249), and the American Lung Association Courtney Cox Cole Lung Cancer Research Award (LCD1421064). We also thank the UTSW Tissue Management Shared Resource, a shared resource at the Simmons Comprehensive Cancer Center, which is supported in part by the National Cancer Institute P30 CA142543. Slide scanning was made possible on Zeiss Axioscan.Z1 and Hamamatsu NanoZoomer S60, courtesy of the following funding (1S10OD032267-01, to Denise Ramirez) and the Whole Brain Imaging Core at UTSW. Small animal imaging was provided by the UTSW Pre-Clinical MRI Core supported by the Cancer Prevention Research Institute of Texas (CPRIT RP210099), the NIH (1S10OD032303-01), and the UTSW Pre-Clinical Radiation Core Facility (PCIRCF). The North Texas Clinical Pharmacology Core is supported by the Cancer Prevention Research Institute of Texas (CPRIT RP210209). TGO was supported by NCI awards U24CA213274 and R01-CA244841-05 and received support as a Duke Science & Technology Scholar.

## Author contributions

**Hari Shankar Sunil**: Data curation; Formal analysis; Validation; Investigation; Visualization; Writing—original draft; Writing—review and editing. **Jean R Clemenceau**: Formal analysis; Visualization; Methodology; Writing—review and editing. **Anthony Grichuk**: Formal analysis; Investigation; Writing—review and editing. **Isabel Barnfather**: Investigation; Writing—review and editing. **Sumanth R Nakkireddy**: Formal analysis; Investigation; Writing—review and editing. **Luke Izzo**: Formal analysis; Investigation; Methodology; Writing—review and editing. **Qiang Feng**: Formal analysis; Investigation; Methodology; Writing—review and editing. **William Hartnett**: Formal analysis; Investigation; Writing—review and editing. **Bret M Evers**: Formal analysis; Writing—review and editing. **Lisa Thomas**: Investigation; Writing—review and editing. **Indhumathy Subramaniyan**: Formal analysis; Investigation; Methodology; Writing—review and editing. **Li Li**: Investigation; Methodology; Writing—review and editing. **William C Putnam**: Investigation; Methodology; Writing—review and editing. **Steven Hepensteil**: Formal analysis; Investigation; Writing—review and editing. **Jingfei Zhu**: Investigation; Writing—review and editing. **Barrett Updegraff**: Investigation; Writing—review and editing. **John D Minna**: Investigation; Writing—review and editing. **Ralph J DeBerardinis**: Investigation; Methodology; Writing—review and editing. **Tae Hyun Hwang**: Formal analysis; Investigation; Methodology; Writing—review and editing. **Jinming Gao**: Investigation; Methodology; Writing—original draft; Writing—review and editing. **Trudy G Oliver**: Investigation; Methodology; Writing—original draft; Writing—review and editing. **Kathryn A O'Donnell**: Conceptualization; Formal analysis; Supervision; Funding acquisition; Visualization; Writing—original draft; Project administration; Writing—review and editing.

Source data underlying figure panels in this paper may have individual authorship assigned. Where available, figure panel/source data authorship is listed in the following database record: biostudies:S-SCDT-10_1038-S44319-025-00631-1.

## Disclosure and competing interests statement

KAO is a scientific co-founder and owns stock in ProtomAb Therapeutics, Inc. The remaining authors declare no competing interests.

# Expanded View Figures

**Figure EV1.  *Tmprss11b* depletion inhibits tumor burden in a syngeneic mouse model of LUSC.**

(**A**) Agarose gel electrophoresis images of PCR amplified products from the Surveyor assay performed on the genomic DNA isolated from KLN205 cells expressing control or *Tmprss11b* sg1 or *Tmprss11b* sg2. The assay was repeated two times with different surveyor primers to confirm the observations (biological replicates). (**B**) Image showing the resected tumors at endpoint from the syngeneic experiment in Fig. 1C. (**C**) qRT-PCR analysis of *Tmprss11b* mRNA in KLN205 cells expressing doxycycline-inducible control shRNA or two independent shRNA sequences targeting *Tmprss11b*. Brown–Forsythe and Welch ANOVA test with Dunnett's T3 multiple comparisons test was used for the statistical analysis, ****$P < 0.0001$. Plot represents mean ± SD; $n = 4$ per group (technical replicates). Experiment was repeated two times for confirmation (biological replicates). (**D**) Quantification of tumor volumes of KLN205 cells expressing doxycycline-inducible control or *Tmprss11b* shRNA on day 50 (terminal measurement) post injection in syngeneic DBA/2 wild-type mice ($n = 10$ control shRNA mice; $n = 8$ *Tmprss11b* sh1 mice; $n = 10$ *Tmprss11b* sh2 mice, biological replicates). Ordinary one-way ANOVA with Dunnett's multiple comparisons test was used for the statistical analysis, $P = 0.0235$. Plot represents mean ± SD. (**E**) Image showing the resected tumors from the syngeneic experiment in Fig. 1D. (**F**) qRT-PCR analysis of *Tmprss11b* mRNA in the resected KLN205 tumors from D). Brown–Forsythe and Welch ANOVA test with Dunnett's T3 multiple comparisons test was used for the statistical analysis, $P = 0.0055$ (*T11b* shRNA1), $P = 0.0041$(*T11b* shRNA2). Plot represents mean ± SD; $n = 3$ technical replicates, $n = 3$-6 tumors per group, biological replicates. (**G**) Quantification of CD8a staining from fluorescent immunohistochemistry (IHC-F) performed on KLN205 tumor sections. An unpaired t test with Welch's correction was used for the statistical analysis ($n = 10$–14 fields per tumor section, 3 tumors per group, biological replicates). Plot represents mean ± SD. Source data are available online for this figure.

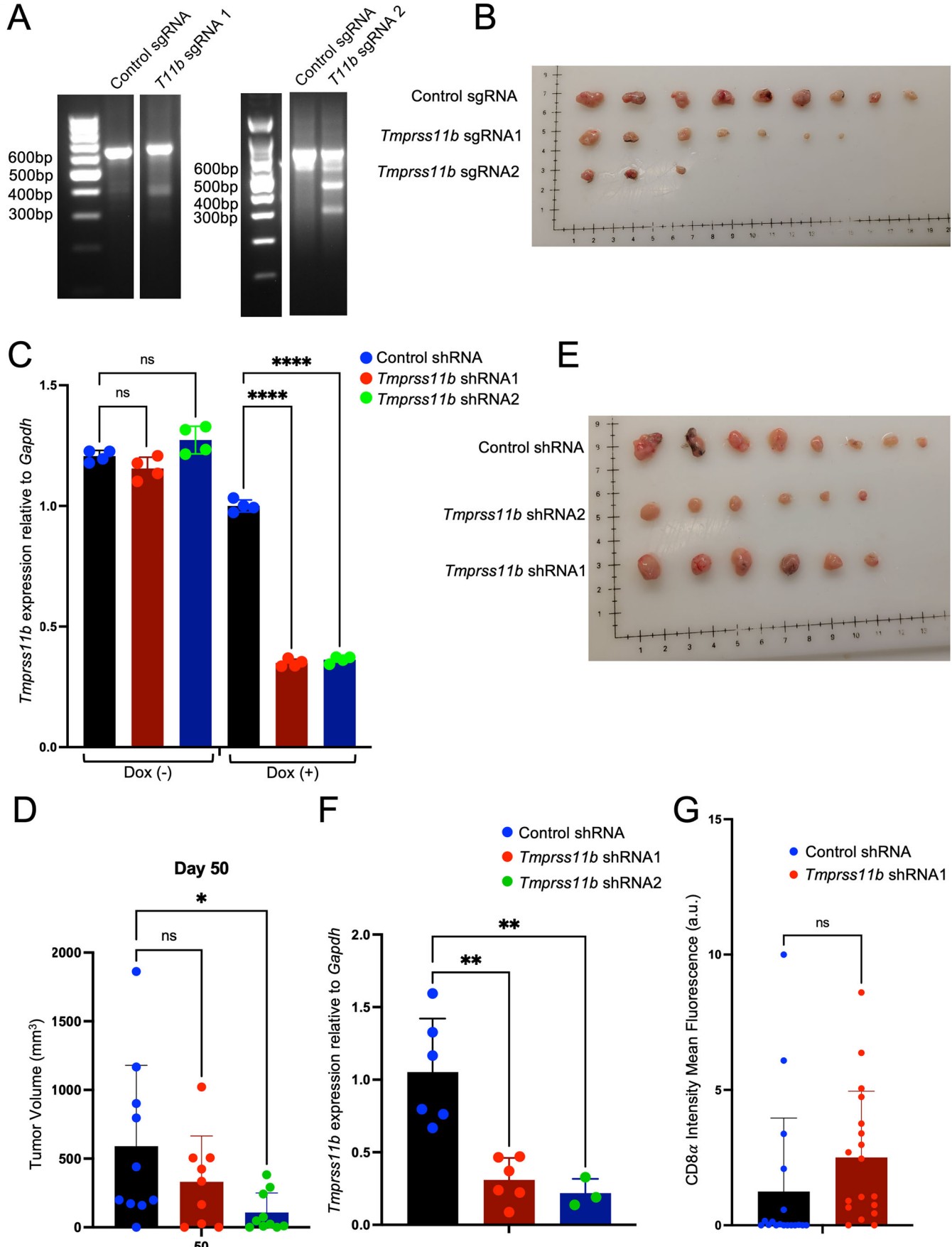

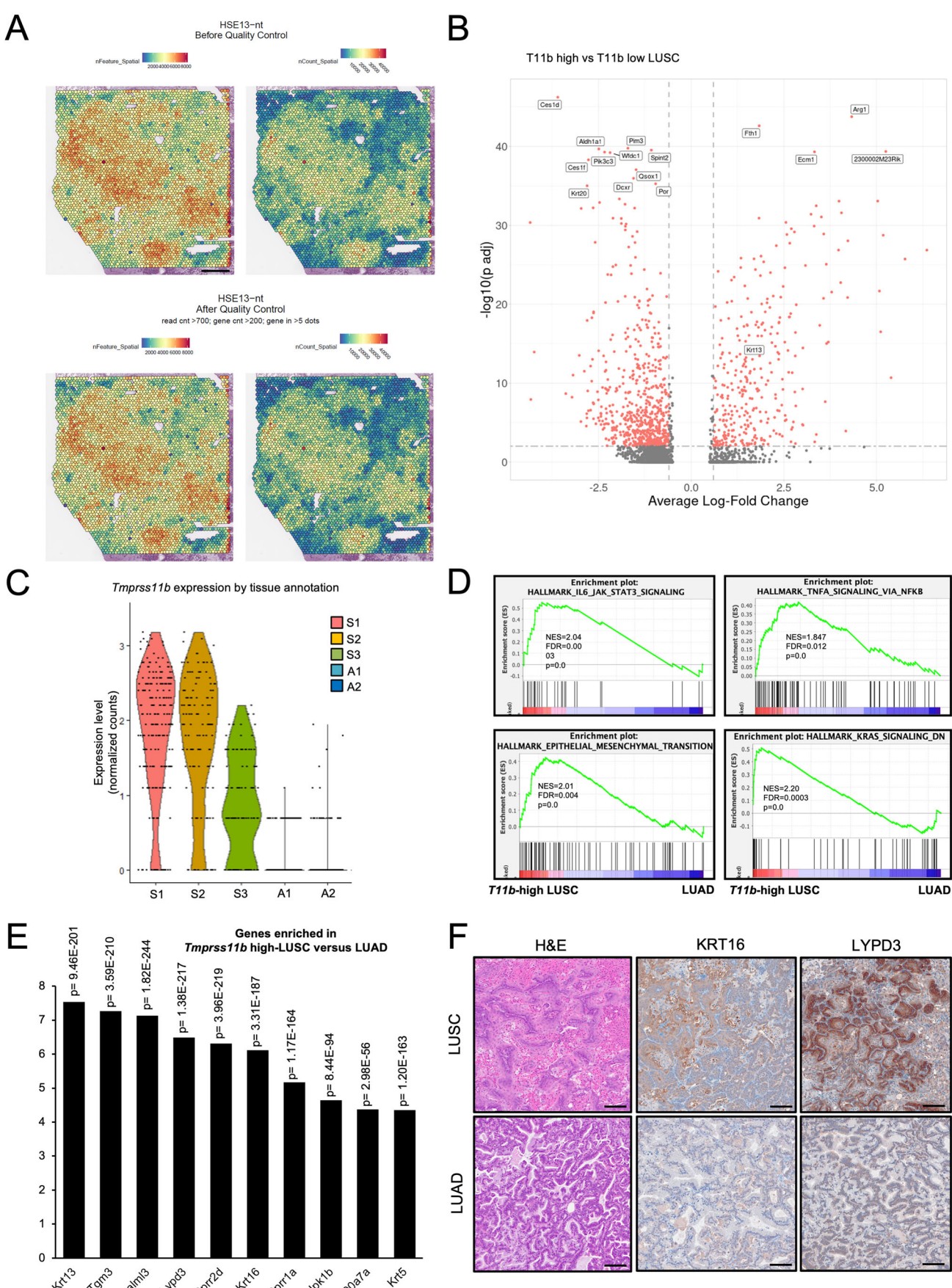

**Figure EV2.**  *Tmprss11b*-high squamous tumors have increased expression of oncogenes and squamous markers.

(A) Spatial plots depicting the read counts before and after the quality control process. Scale bar, 1 mm. (B) Volcano plot showing the top differentially expressed genes in the *Tmprss11b*-high versus low LUSCs spatial data. A total of 4090 genes are plotted, of which 970 pass filter (colored). A two-tailed Wilcoxon rank-sum test with Bonferroni correction was used for the statistical analysis ($n = 4280$ Visium spots post-QC, biological replicates). (C) Violin plot depicting normalized counts for *Tmprss11b* transcript in the annotated regions (from Fig. 2B) ($n = 1597$, biological replicates). (D) Gene set enrichment analysis (GSEA) of the *Tmprss11b*-high LUSC versus LUAD spatial data with normalized enrichment scores (NES), false discovery rate (FDR) and *P* values for the indicated gene signatures. The nominal *P* and FDR values were obtained from the "GSEA Preranked" tool (from Broad Institute) using a weighted scoring scheme. Gene sets were evaluated based on the default normalized enrichment score method, and statistical significance was determined by bootstrapping with 1000 permutations. (E) Top squamous markers and known oncogenes from the differential gene expression (DEG) analysis of the *Tmprss11b*-high LUSC versus LUAD spatial data. Differential gene expression was calculated using Seurat's FindAllMarkers function, and direct comparisons between two classes and corresponding *P* values were obtained using Seurat's FindMarkers function with the Wilcoxon rank-sum test with Bonferroni *P* value correction. (F) Representative H&E and immunohistochemistry (IHC) validation for KRT16 and LYPD3 in LUSC (top) and mucinous LUAD (bottom). The staining was repeated with lung sections from different mice ($n = 2$–3, biological replicates). Scale bar, 100 μm. Source data are available online for this figure

**A**

**Keratin genes showing reduced expression in *Tmprss11b* knockdown KLN205 tumors**

| Gene | *T11b* sh1 log2FC | FDR | *T11b* sh2 log2FC | FDR |
|---|---|---|---|---|
| *Krt1* | -5.13 | 0.195 | -8.66 | 0.015 |
| *Krt5* | -3.51 | 0.353 | -7.48 | 0.015 |
| *Krt6b* | -3.52 | 0.33 | -4.76 | 0.13 |
| *Krt10* | -2.64 | 0.195 | -2.13 | 0.54 |
| *Krt16* | -3.02 | 0.013 | -2.7 | 0.051 |
| *Krtdap* | -3.78 | 0.69 (p=0.0204) | -7.37 | 0.09 |

**B**

**Keratin genes enriched in *T11b*-high versus *T11b*-low LUSCs spatial data**

| Gene | log2FC | Adj p -value |
|---|---|---|
| *Krt16* | 4.03 | 1.85E-23 |
| *Krt6b* | 2.78 | 7.03E-30 |
| *Krt14* | 2.66 | 1.85E-28 |
| *Krtdap* | 2.33 | 5.02E-17 |
| *Krt17* | 1.59 | 3.20E-12 |
| *Krt4* | 1.58 | 5.50E-12 |
| *Krt13* | 1.48 | 5.72E-14 |
| *Krt6a* | 1.405 | 6.74E-13 |

**C**

**Keratin genes enriched in *T11b*-high LUSCs versus LUADs spatial data**

| Gene | log2FC | Adj p -value |
|---|---|---|
| *Krt13* | 7.53 | 9.46E-201 |
| *Krtdap* | 7.39 | 1.07E-231 |
| *Krt6b* | 7.29 | 7.74E-227 |
| *Krt16* | 6.12 | 3.31E-187 |
| *Krt6a* | 5.71 | 1.07E-190 |
| *Krt14* | 5.55 | 1.21E-185 |
| *Krt17* | 5.25 | 1.39E-197 |
| *Krt5* | 4.35 | 1.20E-163 |

**D**

**Keratin genes enriched in *T11B*-high versus *T11B*-low LUSCs TCGA data**

| Gene | log ratio | T-test p -value |
|---|---|---|
| *KRT13* | 4.85 | 1.30715E-33 |
| *KRT24* | 4.17 | 1.47289E-19 |
| *KRTDAP* | 3.96 | 4.58479E-20 |
| *KRT78* | 3.44 | 1.44602E-31 |
| *KRT16* | 2.92 | 2.58855E-27 |
| *KRT6C* | 2.83 | 4.80952E-33 |
| *KRT10* | 2.81 | 1.24801E-11 |
| *KRT6B* | 2.79 | 3.89382E-31 |
| *KRT14* | 2.69 | 2.48345E-12 |
| *KRT6A* | 2.50 | 3.68511E-28 |

**E**

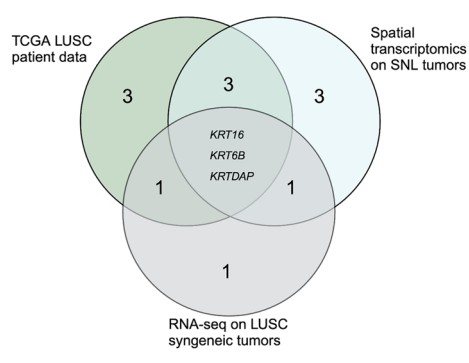

**Figure EV3.   Modulation of TMPRSS11B expression is accompanied by coordinated changes in expression of Keratin genes.**

(A) Top downregulated Keratin genes from differential gene expression analysis of control shRNA versus *Tmprss11b* shRNA bulk RNA sequencing from the KLN205 syngeneic experiment in Fig. 1D. The log2FC change depicts the reduction in expression of the indicated genes in the *Tmprss11b* knockdown tumors compared to the control. (B) Top Keratin genes from the differential gene expression (DEG) analysis of the *Tmprss11b*-high versus low in LUSC spatial data from SNL lung tumors. (C) Top keratin genes from the differential gene expression (DEG) analysis of the *Tmprss11b*-high LUSC versus LUAD spatial data from SNL lung tumors. (D) Top Keratin genes from the differential gene expression (DEG) analysis of *TMPRSS11B*-high versus low LUSC human tumors from TCGA. (E) Venn diagram depicting overlapping Keratin genes from the gene lists in (A–D). Source data are available online for this figure

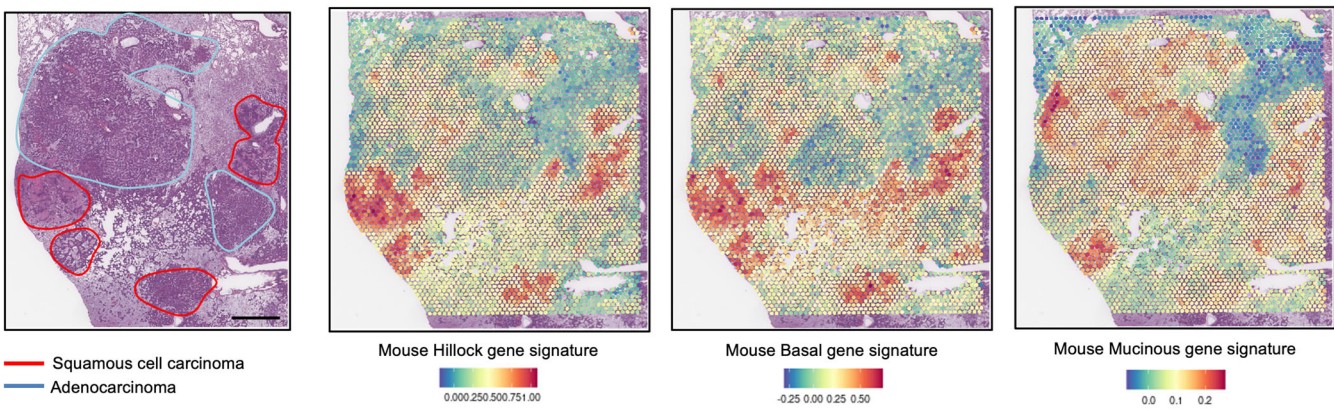

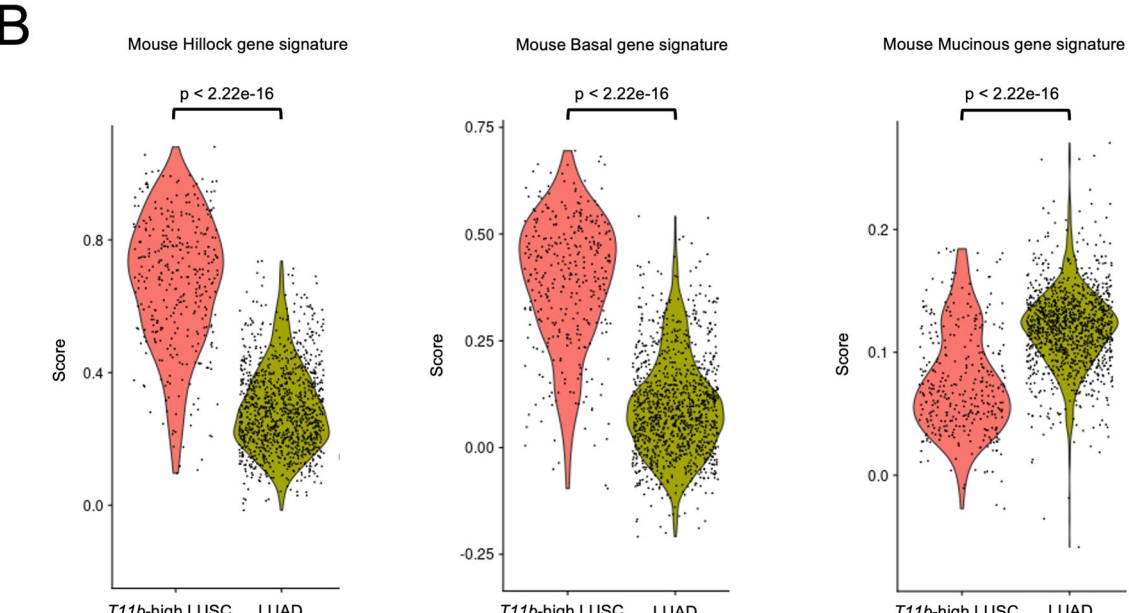

**Figure EV4. *Tmprss11b*-high squamous tumors show enrichment for hillock and basal gene signatures.**

(A) H&E image of the lung section from 3 A), annotated with regions of LUSC and LUAD, (left) and spatial plots from the transcriptomic data depicting the distribution of the indicated gene signatures (right). Scale bar, 1 mm. (B) Violin plots representing the enrichment for the indicated gene signatures in *Tmprss11b*-high LUSC and LUAD. A two-tailed Wilcoxon rank-sum test was used for the statistical analysis (*n* = 1492, biological replicates). *P* < 2.22e-16 for all comparisons. Source data are available online for this figure

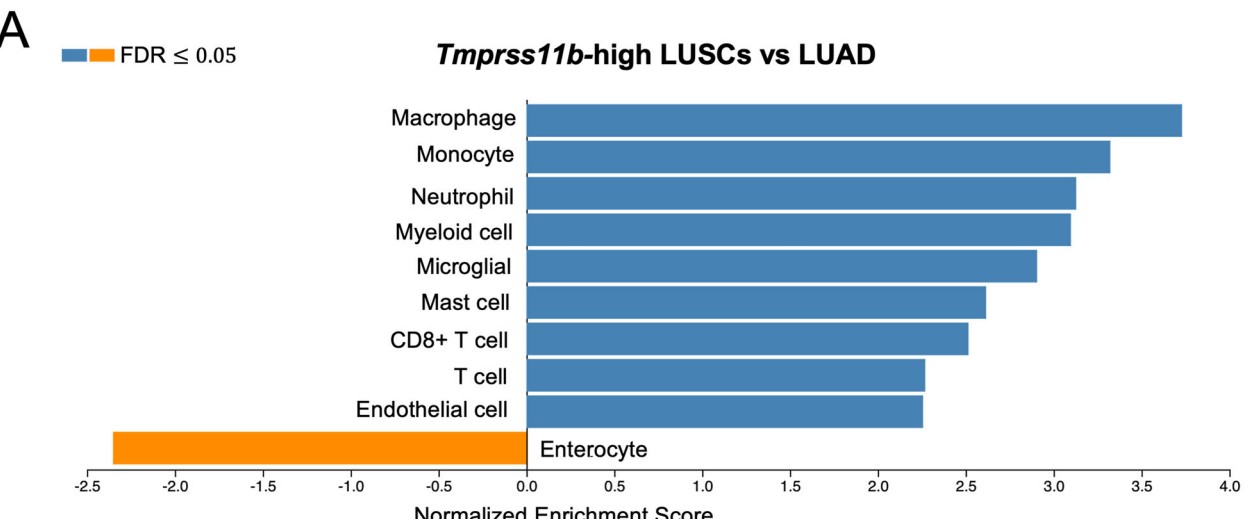

A

FDR ≤ 0.05

**Tmprss11b-high LUSCs vs LUAD**

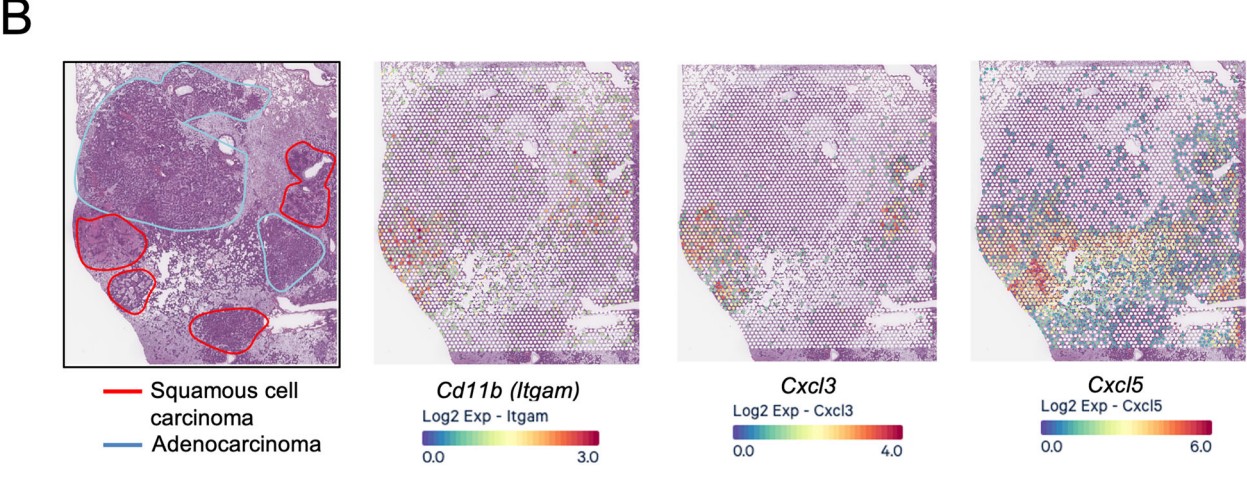

B

Squamous cell carcinoma
Adenocarcinoma

*Cd11b (Itgam)*
Log2 Exp - Itgam
0.0   3.0

*Cxcl3*
Log2 Exp - Cxcl3
0.0   4.0

*Cxcl5*
Log2 Exp - Cxcl5
0.0   6.0

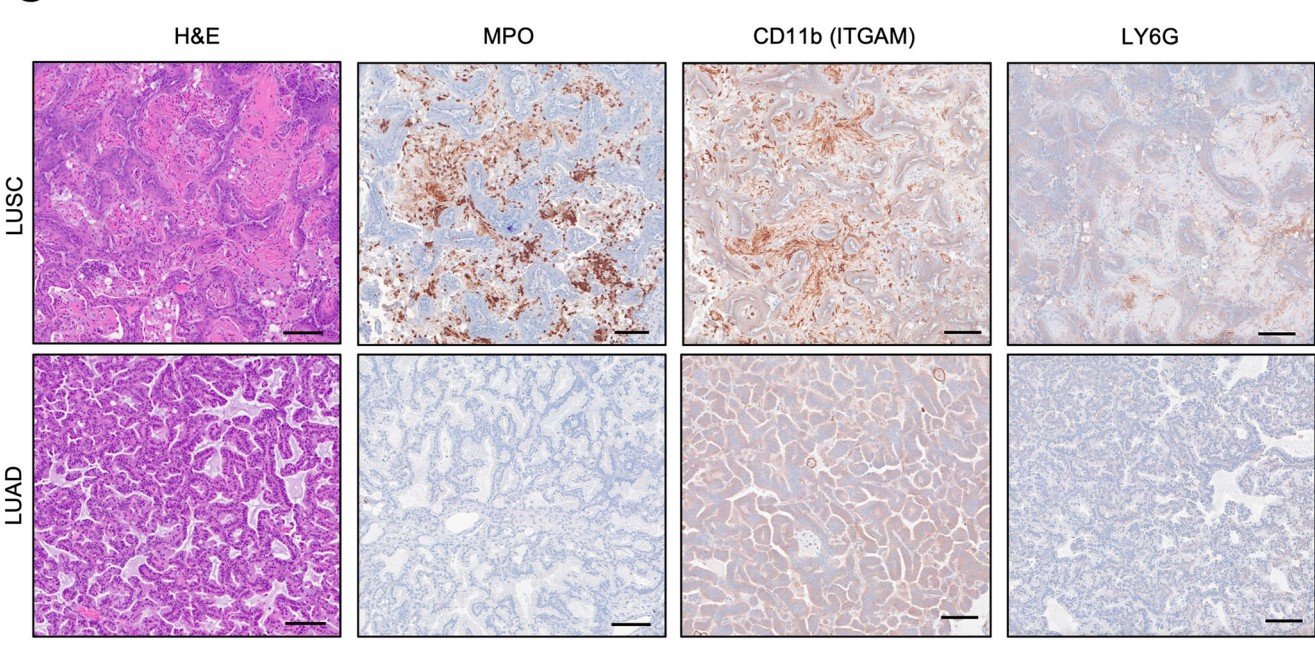

C

H&E    MPO    CD11b (ITGAM)    LY6G

LUSC

LUAD

**Figure EV5. *Tmprss11b*-high squamous tumors have higher infiltration of neutrophils.**

(A) Gene set enrichment analysis (GSEA) of *Tmprss11b*-high LUSC versus LUAD spatial transcriptomics data with normalized enrichment scores (NES) and false discovery rate (FDR) for the indicated gene signatures; analysis performed using WEB-based Gene Set Analysis Toolkit. (B) H&E image of the lung section from Fig. 3A), annotated with regions of LUSC and LUAD, and corresponding spatial plots depicting the distribution of the indicated mRNAs (neutrophil markers). (C) Representative H&E and immunohistochemistry (IHC) validation for MPO, CD11b (ITGAM) and LY6G in LUSC (top) and mucinous LUAD (bottom). The staining was repeated with lung sections from different mice ($n = 2$, biological replicates). Scale bar, 100 μm. Source data are available online for this figure

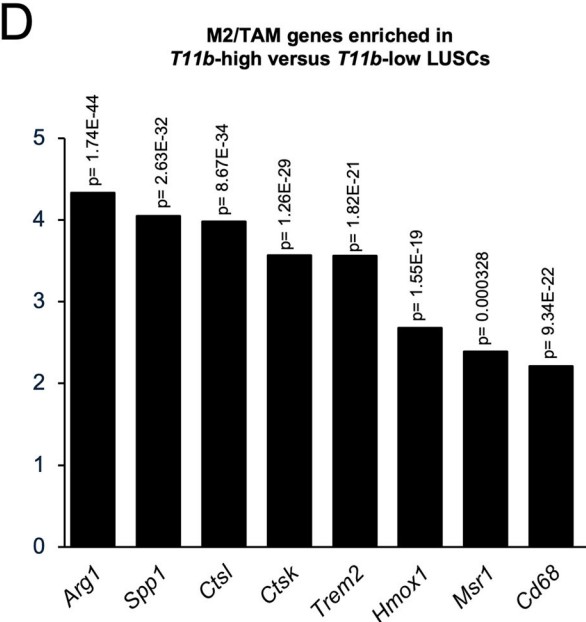

◀ **Figure EV6.** *Tmprss11b*-high squamous tumors show enrichment for M2-like macrophage markers.

(A) Quantification of immune cell populations in *Tmprss11b*-high LUSC vs. LUAD using cell deconvolution analysis of the spatial data. A two-sided Mann–Whitney–Wilcoxon test was used for the statistical analysis (*T11b*-high LUSC n = 344, LUAD n = 1148, biological replicates), ****P = 7.96E-78 (Classical monocyte), ****P = 7.63E-16 (T cell), ****P = 4.74E-17 (Alveolar macrophage), ****P = 6.04E-133 (Leukocyte), ***P = 2.31E-05 (Non-classical monocyte), ****P = 2.84E-10 (Lung endothelial cell), ****P = 1.28E-111 (Stromal cell), ****P = 3.92E-09 (B cell), ****P = 4.41E-44 (Ciliated columnar cell of tracheobronchial tree). The box plots represent the distribution of values for each group, extending from the 25th percentile to the 75th percentile; *T11b*-high LUSC: Classical monocyte (minima=0.00035, median=0.32265, maxima=1, Q1 = 0.16193, Q3 = 0.52806), Natural killer cell (minima=0.04763, median=0.37114, maxima=0.82722, Q1 = 0.28091, Q3 = 0.49943), T cell (minima=0, median=0.18974, maxima=0.72977, Q1 = 0.06047, Q3 = 0.32819), Alveolar macrophages (minima=0.01914, median=0.44117, maxima=0.91769, Q1 = 0.32846, Q3 = 0.56415), Leukocyte (minima=0.9.87E-05, median=0.41385, maxima=1, Q1 = 0.27138, Q3 = 0.61336), Myeloid cell (minima=0.00012, median=0.04951, maxima=0.61895, Q1 = 0.00328, Q3 = 0.24955), Non-classical monocyte (minima=0.00036, median=0.25014, maxima=0.92007, Q1 = 0.13621, Q3 = 0.44976), Lung endothelial cell (minima=0, median=0.18464, maxima=0.52498, Q1 = 0.10807, Q3 = 0.27484), Stromal cell (minima=0, median=0.44905, maxima=0.94097, Q1 = 0.31863, Q3 = 0.56757), B cell (minima=0, median=0.24525, maxima=0.69056, Q1 = 0.13767, Q3 = 0.35883), Mast cell (minima=0.00046, median=0.09658, maxima=0.30038, Q1 = 0.06603, Q3 = 0.15977), Ciliated columnar cell of tracheobronchial tree (minima=0.05124, median=0.18792, maxima=0.34003, Q1 = 0.15953, Q3 = 0.23173), Type II pneumocyte (minima=0, median=0.28766, maxima=0.71305, Q1 = 0.19541, Q3 = 0.40246); LUAD: Classical monocyte (minima=0, median=0.00788, maxima=0.47288, Q1 = 0.00108, Q3 = 0.18980), Natural killer cell (minima=0.04933, median=0.39471, maxima=0.76680, Q1 = 0.31838, Q3 = 0.49774), T cell (minima=0.00029, median=0.28494, maxima=0.70883, Q1 = 0.19111, Q3 = 0.39820), Alveolar macrophages (minima=0.04437, median=0.36127, maxima=0.67967, Q1 = 0.28261, Q3 = 0.44143), Leukocyte (minima=0, median=0.00264, maxima=0.33064, Q1 = 0.00058, Q3 = 0.13260), Myeloid cell (minima=0, median=0.05014, maxima=0.64748, Q1 = 0.00252, Q3 = 0.26050), Non-classical monocyte (minima=0, median=0.22165, maxima=0.68314, Q1 = 0.10763, Q3 = 0.33783), Lung endothelial cell (minima=0, median=0.23560, maxima=0.51640, Q1 = 0.16346, Q3 = 0.30464), Stromal cell (minima=0.39036, median=0.75595, maxima=1, Q1 = 0.66414, Q3 = 0.84667), B cell (minima=0, median=0.30153, maxima=0.62486, Q1 = 0.22758, Q3 = 0.38650), Mast cell (minima=0, median=0.08179, maxima=0.20366, Q1 = 0.06479, Q3 = 0.12034), Ciliated columnar cell of tracheobronchial tree (minima=0.08851, median=0.25076, maxima=0.41768, Q1 = 0.21195, Q3 = 0.29424), Type II pneumocyte (minima=0, median=0.29401, maxima=0.63087, Q1 = 0.21685, Q3 = 0.38246). (B) Quantification of immune cell populations in *Tmprss11b*-high vs. *Tmprss11b*-low LUSC using cell deconvolution analysis of the spatial data. A two-sided Mann–Whitney-Wilcoxon test was used for the statistical analysis (*T11b*-high LUSC n = 344, *Tmprss11b*-low LUSC n = 104, biological replicates), ****P = 8.89E-08 (Classical monocyte), ****P = 3.22E-15 (Alveolar macrophage), ****P = 1.25E-32 (Leukocyte), ***P = 6.54E-05 (Non-classical monocyte), ****P = 2.42E-36 (Ciliated columnar cell of tracheobronchial tree). The box plots represent the distribution of values for each group, extending from the 25th percentile to the 75th percentile; *T11b*-high LUSC: Classical monocyte (minima=0.00029, median=0.30549, maxima=1, Q1 = 0.15744, Q3 = 0.49527), Natural killer cell (minima=0, median=0.41140, maxima=0.97370, Q1 = 0.26653, Q3 = 0.5490), T cell (minima=0, median=0.18983, maxima=0.81820, Q1 = 0.05084, Q3 = 0.35778), Alveolar macrophages (minima=0.00293, median=0.43913, maxima=0.91408, Q1 = 0.34461, Q3 = 0.57240), Leukocyte (minima=0.00068, median=0.40974, maxima=1, Q1 = 0.26157, Q3 = 0.62635), Myeloid cell (minima=4.55E-05, median=0.09014, maxima=0.65331, Q1 = 0.00217, Q3 = 0.26263), Non-classical monocyte (minima=1.67E-05, median=0.26527, maxima=0.85779, Q1 = 0.14324, Q3 = 0.42906), Lung endothelial cell (minima=0, median=0.17896, maxima=0.55746, Q1 = 0.09927, Q3 = 0.28255), Stromal cell (minima=0, median=0.44430, maxima=1, Q1 = 0.26373, Q3 = 0.62252), B cell (minima=0, median=0.24471, maxima=0.69114, Q1 = 0.14794, Q3 = 0.36522), Mast cell (minima=0, median=0.10069, maxima=0.30702, Q1 = 0.06993, Q3 = 0.16477), Ciliated columnar cell of tracheobronchial tree (minima=0.03638, median=0.19021, maxima=0.35185, Q1 = 0.15468, Q3 = 0.23355), Type II pneumocyte (minima=0, median=0.24117, maxima=0.67966, Q1 = 0.1531, Q3 = 0.36374); *Tmprss11b*-low LUSC: Classical monocyte (minima=0.00020, median=0.19138, maxima=0.60530, Q1 = 0.04595, Q3 = 0.26969), Natural killer cell (minima=0, median=0.37341, maxima=0.84796, Q1 = 0.29730, Q3 = 0.51756), T cell (minima=7.77E-05, median=0.24393, maxima=0.74889, Q1 = 0.10171, Q3 = 0.36058), Alveolar macrophages (minima=0.08051, median=0.30265, maxima=0.60637, Q1 = 0.23290, Q3 = 0.38229), Leukocyte (minima=0, median=0.10504, maxima=0.43837, Q1 = 0.03357, Q3 = 0.19549), Myeloid cell (minima=0.00048, median=0.20803, maxima=0.90453, Q1 = 0.00571, Q3 = 0.36524), Non-classical monocyte (minima=0.00013, median=0.19676, maxima=0.60907, Q1 = 0.08942, Q3 = 0.29728), Lung endothelial cell (minima=0, median=0.18352, maxima=0.39994, Q1 = 0.10900, Q3 = 0.23955), Stromal cell (minima=0.10516, median=0.53043, maxima=0.88675, Q1 = 0.32928, Q3 = 0.69341), B cell (minima=0.00077, median=0.28140, maxima=0.64062, Q1 = 0.17045, Q3 = 0.35852), Mast cell (minima=0, median=0.09511, maxima=0.21496, Q1 = 0.06911, Q3 = 0.12745), Ciliated columnar cell of tracheobronchial tree (minima=0.13684, median=0.37548, maxima=0.66299, Q1 = 0.30862, Q3 = 0.45037), Type II pneumocyte (minima=0.06470, median=0.26157, maxima=0.47461, Q1 = 0.21841, Q3 = 0.32089). (C) Gene set enrichment analysis (GSEA) of the *Tmprss11b*-high versus low LUSC spatial data with normalized enrichment scores (NES), false discovery rate (FDR) and p values for the indicated immune cell gene signatures. The nominal *P* and FDR values were obtained from the "GSEA Preranked" tool (from Broad Institute) using a weighted scoring scheme. Gene sets were evaluated based on the default normalized enrichment score method, and statistical significance was determined by bootstrapping with 1000 permutations. (D) Bar graph representing top M2-like/TAM genes from the differential gene expression (DEG) analysis of *Tmprss11b*-high versus low LUSC spatial transcriptomics data. Differential gene expression was calculated using Seurat's FindAllMarkers function, and direct comparisons between two classes and corresponding *P* values were obtained using Seurat's FindMarkers function with the Wilcoxon rank-sum test with Bonferroni *P* value correction. Source data are available online for this figure

