## [Peer Review File · EMBO Reports]

TMPRSS11B promotes an acidified microenvironment and immune suppression in squamous lung cancer

Hari Sunil, Jean Clemenceau, Anthony Grichuk, Isabel Barnfather, Sumanth Nakkireddy, Luke Izzo, Qiang Feng, William Hartnett, Bret Evers, Lisa Thomas, Indhumathy Subramaniyan, Li Li, William Putnam, Steven Hepenstein, Jingfei Zhu, Barrett Updegraff, John Minna, Ralph DeBerardinis, Tae Hwang, Jinming Gao, Trudy Oliver, and Kathryn O'Donnell

Corresponding author(s): Kathryn O'Donnell (Kathryn.ODonnell@UTSouthwestern.edu)

Review Timeline:

Submission Date:	14th Apr 25
Editorial Decision:	23rd May 25
Revision Received:	21st Sep 25
Editorial Decision:	7th Oct 25
Revision Received:	20th Oct 25
Accepted:	24th Oct 25

Editor: Achim Breiling

Transaction Report:

Dear Dr. O'Donnell,

Thank you for the submission of your manuscript to EMBO reports. I have now received reports from two of the three referees that were asked to evaluate your study, which can be found at the end of this email. A third referee had agreed to assess the manuscript, but never submitted a report and was completely unresponsive to our chasers. I thus decided to proceed without his/her input.

As you will see, the remaining two referees think that these findings are of interest. However, they have several comments, concerns, and suggestions, indicating that a major revision of the manuscript is necessary to allow publication of the study in EMBO reports. As the reports are below, and all the referee concerns need to be addressed, I will not detail them here.

Given the constructive referee comments, I would like to invite you to revise your manuscript with the understanding that the concerns of the referees must be addressed in the revised manuscript and/or in a detailed point-by-point response. Acceptance of your manuscript will depend on a positive outcome of a second round of review. It is EMBO reports policy to allow a single round of revision only and acceptance of the manuscript will therefore depend on the completeness of your responses included in the next, final version of the manuscript.

- 1) a .docx formatted version of the final manuscript text (including legends for main figures, EV figures and tables), but without the figures included. Figure legends should be compiled at the end of the manuscript text.
- 2) individual production quality figure files as .eps, .tif, .jpg (one file per figure), of main figures and EV figures. Please upload these as separate, individual files upon re-submission.

- 4) a complete author checklist, which you can download from our author guidelines

(<https://www.embopress.org/page/journal/14693178/authorguide>). Please insert page numbers in the checklist to indicate where the requested information can be found in the manuscript. The completed author checklist will also be part of the RPF.

5) that primary datasets produced in this study (e.g. RNA-seq, CHIP-seq, structural and array data) are deposited in an appropriate public database. If no primary datasets have been deposited, please also state this in a dedicated section (e.g. 'No primary datasets have been generated and deposited'), see below.

The accession numbers and database should be listed in a formal "Data Availability" section that follows the model below. This is now mandatory (like the COI statement). Please note that the Data Availability Section is restricted to new primary data that are part of this study. This section is mandatory. As indicated above, if no primary datasets have been deposited, please state this in this section

Data availability

6) We now request the publication of original source data with the aim of making primary data more accessible and transparent to the reader. You will receive a separate email with instructions for providing source data with your revised manuscript, including information how to upload and organize the files.

8) Regarding data quantification and statistics, please make sure that the number "n" for how many independent experiments were performed, their nature (biological versus technical replicates), the bars and error bars (e.g. SEM, SD) and the test used to calculate p-values is indicated in the respective figure legends (also for EV and Appendix figures). Please also check that all the p-values are explained in the legend, and that these fit to those shown in the figure. Please provide statistical testing where applicable. Please avoid the phrase 'independent experiment', but clearly state if these were biological or technical replicates. Please also indicate (e.g. with n.s.) if testing was performed, but the differences are not significant. In case n=2, please show the data as separate datapoints without error bars and statistics. See also: <http://www.embopress.org/page/journal/14693178/authorguide#statisticalanalysis>

9) Please add scale bars of similar style and thickness to all microscopic images, using clearly visible black or white bars (depending on the background). Please place these in the lower right corner of the images themselves. Please do not write on or near the bars in the image but define the size in the respective figure legend.

10) Please also note our reference format:

12) We now use CRediT to specify the contributions of each author in the journal submission system. CRediT replaces the author contribution section. Please use the free text box to provide more detailed descriptions and do NOT provide your final manuscript text file with an author contributions section. See also our guide to authors: <https://www.embopress.org/page/journal/14693178/authorguide#authorshipguidelines>

13) All Materials and Methods need to be described in the main text using our 'Structured Methods' format, which is required for all research articles. According to this format, the Methods section should include a Reagents and Tools Table (listing key reagents, experimental models, software, and relevant equipment and including their sources and relevant identifiers), uploaded as separate file, and a Methods section in which we encourage the authors to describe their methods using a step-by-step protocol format with bullet points, to facilitate the adoption of the methodologies across labs. More information on how to adhere to this format as well as downloadable templates (.doc) for the Reagents and Tools Table can be found in our author guidelines (section 'Structured Methods'):

14) Please reduce the keywords to five and order the sections like this, using these names:

Title page - Abstract - Keywords - Introduction - Results - Discussion - Methods - Data availability section - Acknowledgements - Disclosure and Competing Interests Statement - References - Figure legends - Expanded View Figure legends

15) Please make sure that all the funding information is also entered into the online submission system and that it is complete and similar to the one in the acknowledgement section of the manuscript text file.

I look forward to seeing a revised form of your manuscript when it is ready.

Yours sincerely,

Referee #1:

In their manuscript, Sunil and colleagues report the importance of the transmembrane serine protease, TMPRSS11B, in the progression of lung squamous cell carcinoma (LUSC). They used an immunocompetent syngeneic model of subcutaneous transplantation of a LUSC cell line after TMPRSS11B deletion (or not). The approaches for loss-of-function consisted of constitutive knockout through CRISPR/Cas9 and doxycycline-inducible shRNA-based knockdown. An increased presence of T lymphocytes characterised the KO tumors, which grew less efficiently and had fewer pERK staining, a mark of tumor progression. Next, the authors used the autochthonous SNL model, and performed deep comparative analyses of LUAD to LUSC histological regions, showing enrichment of TMPRSS11B in the latter. Combining and comparing multiple spatial resolution methodologies, they identified TMPRSS11B high squamous regions linked to local acidification and increased presence of M2-like macrophages. Although this part is mostly correlative, altogether the amount of work performed by the authors is impressive, well executed and provides new insights into the role of this protease and associated mechanisms of action in LUSC development.

I have a few comments:

1) In vivo, the concept of "M1" and "M2" like macrophages in tumors is decreasing in importance and relevance with new molecular data. Some discussion about that could be added. Also, reviewer wonders how other markers of tumor-associated macrophage polarisation would be expressed in the regions studied by the authors. According to (PMID: 37535729), it could be interesting to specifically monitor the expression of CXCL9 and SPP1. Do they differ in TAMs from TMPRSS11B high vs low regions, or LUAD vs LUSC?

2) Could authors stain tumors +/- TMPRSS11B deletion (collected for Fig 1F) with macrophage polarity markers, such as those used in Fig. 6F (HMOX1, MSR1), CXCL9 and SPP1? This could help substantiate the conclusion of TMPRSS11B affecting TAM

polarisation.

3) It would be nice, at least in vitro, to demonstrate the reduction of TMPRSS11B protein levels upon Dox-induced shRNA expression. At the same time, additional controls should be included = shRNA1 and shRNA2 without Dox to compare to control shRNA with or without Dox, to show that the system is not leaky.

4) Fig. 3B and 4A. Region S3 seems to display intermediate TMPRSS11B expression. Could this be a tumor with an adeno-to-squamous transition? To investigate this, a staining with KRT16 and LYPD3 could help, as performed in Fig. S2F.

Referee #2:

1. Does this manuscript report a single key finding?

YES

LUSC tumors with elevated TMPRSS11B, a trypsin-like serine protease that promotes MCT4-mediated lactate export, exhibit a tumor microenvironment depleted of CD4/CD8 T cells and enriched with immunosuppressive M2-like macrophages.

2. Is the reported work of significance (YES), or does it describe a confirmatory finding or one that has already been documented using other methods or in other organisms etc (NO)?

YES

This study investigates the role of TMPRSS11B in shaping the tumor microenvironment (TME) of LUSC tumors using immunocompetent mouse models, an area not previously published.

3. Is it of general interest to the molecular biology community? YES/NO

If YES, please say why, in a single sentence. If NO, please state which more specialized community you feel it is aimed at (or none), in a single word or phrase.

NO

This would be of greater interest to the cancer research community.

4. Is the single major finding robustly documented using independent lines of experimental evidence (YES), or is it really just a preliminary report requiring significant further data to become convincing, and thus more suited to a longer-format article (NO)?

YES

The authors present a manuscript examining the role of Tmprss11b in regulating the tumor microenvironment (TME) in lung squamous cell carcinoma (LUSC), a subtype of NSCLC with limited effective therapeutic options. This study employs several immunocompetent mouse models of LUSC to demonstrate that: (a) loss of Tmprss11b slows tumor growth and increases CD4/CD8 T-cell infiltration; (b) spatial transcriptomics reveals the highest Tmprss11b expression in Krt13+ hillock-like cells; (c) spatial transcriptomics indicates enrichment of M2 macrophages in Tmprss11b-expressing LUSC tumors; and (d) metabolomics data show elevated lactate levels in LUSC, with higher lactate in areas of lower pH compared to areas with higher pH. Overall, this well-written manuscript effectively leverages immunocompetent mouse models of LUSC and spatial transcriptomics to characterize the TME in Tmprss11b-high LUSC, Tmprss11b-low LUSC, and LUAD tumors.

Major concerns:

1) Figure 6C shows the accumulation of the pH-sensitive probe (USP 5.3 ICG). The authors argue that ICG accumulates in regions adjacent to Tmprss11b-high LUSC tumors; however, Figure 6C shows ICG-positive areas that are not directly adjacent to Tmprss11b-high LUSC tumors, with ICG accumulating in patches rather than surrounding Tmprss11b-high LUSC tumors. How do the authors reconcile this discrepancy? Could the use of the MCT1 inhibitor AZD3965 or sodium lactate, as described in reference 70, help address this question?

2) Figure 7D compares lactate levels in high-pH versus low-pH areas, but based on Figure S7, it appears that the "high-pH" area primarily consists of normal tissue, while the "low-pH" area is tumor tissue (unspecified as LUSC or LUAD). Can the authors confirm if this interpretation is correct? Similarly, for Figures 6D and 6E, how were the "low-pH" and "high-pH" areas defined?

3) Figures 5B, S5B, S5C, and S6D present quantification of various immune cell populations, including "leukocytes," which are enriched in Tmprss11b-high LUSC. Are "leukocytes" defined as CD45-positive cells? It would be helpful to validate some of the transcriptomics results for immune cells using IHC staining.

4) Neutrophils have been reported to be overrepresented in LUSC compared to LUAD patient samples (PMID: 36368318), yet neutrophils are not present in the Figure S5B, S5C. S6D. It would be helpful to include neutrophils in this dataset.

5) Do Tmprss11b-high LUSC tumors exhibit elevated levels of Glut1, MCT1, and MCT4?

Minor concerns:

1) Some of author affiliations appear (possibly) incorrect.

2) Figure 3E, 5D, 6E, S2E, S5E - I would suggest re-naming this to "Genes enriched in..."

3) Line 540 - "24-hours post-infection" should be "24-hours post-injection"?

- 4) Supplementary Table 1 lists PDL1, FOXP3, CD25 antibodies that were used in IHC, but there are no Figures that include PDL1, FOXP3, CD25.
- 5) Supplementary Table 4 lists metabolites identified (16 total), but only lactate was included in the Figures.

Kathryn A. O'Donnell, Ph.D.
Associate Professor
Department of Molecular Biology
UT Southwestern Medical Center
Dallas, Texas 75390-9148

UT Southwestern
Medical Center

September 19th, 2025

We would like to thank the editor and the reviewers for your favorable responses to our work and for your constructive comments. We have carefully considered each point that was raised, performed an extensive series of new experiments, and modified the text accordingly. Thus, we believe that the manuscript is significantly improved as a result of the review process. We hope that you will agree and deem the revised manuscript suitable for publication in *EMBO Reports*. Our point-by-point responses are provided below.

POINT-BY-POINT RESPONSE TO REVIEWERS

Referee #1:

In their manuscript, Sunil and colleagues report the importance of the transmembrane serine protease, TMPRSS11B, in the progression of lung squamous cell carcinoma (LUSC). They used an immunocompetent syngeneic model of subcutaneous transplantation of a LUSC cell line after TMPRSS11B deletion (or not). The approaches for loss-of-function consisted of constitutive knockout through CRISPR/Cas9 and doxycycline-inducible shRNA-based knockdown. An increased presence of T lymphocytes characterized the KO tumors, which grew less efficiently and had fewer pERK staining, a mark of tumor progression. Next, the authors used the autochthonous SNL model and performed deep comparative analyses of LUAD to LUSC histological regions, showing enrichment of TMPRSS11B in the latter. Combining and comparing multiple spatial resolution methodologies, they identified TMPRSS11B high squamous regions linked to local acidification and increased presence of M2-like macrophages. Although this part is mostly correlative, altogether the amount of work performed by the authors is impressive, well executed and provides new insights into the role of this protease and associated mechanisms of action in LUSC development.

Comments:

1) In vivo, the concept of "M1" and "M2" like macrophages in tumors is decreasing in importance and relevance with new molecular data. Some discussion about that could be added. Also, reviewer wonders how other markers of tumor-associated macrophage polarisation would be expressed in the regions studied by the authors. According to (PMID: 37535729), it could be interesting to specifically monitor the expression of CXCL9 and SPP1. Do they differ in TAMs from TMPRSS11B high vs low regions, or LUAD vs LUSC?

We thank the reviewer for these strongly supportive comments. We agree that the concept of strict “M1” and “M2” terminology may be decreasing in importance because there is a growing appreciation that additional macrophage subsets exist in different tissues and tumor types. We were careful to use the term “M2-like” in our initial submission to acknowledge this, and we emphasize the fact that these macrophages are immunosuppressive, and tumor promoting in the revised manuscript. As requested, we have revised the introduction and discussion accordingly and acknowledge the existence of a continuum of macrophage subtypes in the tumor microenvironment (TME) beyond the M1 and M2 subtypes in the discussion on **pp.10-11**.

Interestingly, we observed a significant enrichment of several tumor-associated macrophage (TAM) polarization genes including *Arg1*, *Spp1*, *Trem2*, *Hmox1*, *C3ar1*, *Msr1*, and *Mmp12*, in *Tmprss11b*-high LUSCs compared to *Tmprss11b*-low LUSCs and LUADs. To the reviewer’s point, we did not see enrichment for several M2 polarization genes such as *Cd163*, *Ii10* and *Stab1*. This is consistent with the observation that macrophages can display a mixed phenotype, rather than strictly fitting into the M1 or M2 subtypes. Interestingly, the TAMs expressed in *Tmprss11b*-high LUSCs share similarities to the lipid associated “LA”-TAMs and immune regulatory “Reg”-TAMs recently described in PMID: 35690521. This is quite intriguing, but we hope the editor and reviewer agree that additional characterization of TAM subtypes is beyond the scope of the current study. In the future, we plan to further characterize the relationship between *TMPRSS11B* expression and TAM subsets using spatial analyses with single cell resolution.

To directly address the reviewer’s comment, we assessed *Cxcl9* and *Spp1* expression in these tumors. These data are provided below and in **Figure 6C** of the revised manuscript. Given that *SPP1* and *CXCL9* are secreted proteins, we reasoned that transcript levels are better suited to capture gene expression differences in the different tumor types in this model. From our spatial dataset, we observed a significant enrichment for *Spp1* transcript in *Tmprss11b*-high LUSCs compared to *Tmprss11b*-low LUSCs and LUADs. In contrast, *Cxcl9* transcripts showed similar, albeit low levels of expression in the different tumor types.

2) Could authors stain tumors +/- TMPRSS11B deletion (collected for Fig 1F) with macrophage polarity markers, such as those used in Fig. 6F (HMOX1, MSR1), CXCL9 and SPP1? This could help substantiate the conclusion of TMPRSS11B affecting TAM polarization.

As requested, we performed IHCs for HMOX1, MSR1, ARG1 and SPP1 in the KLN205 syngeneic tumors +/- TMPRSS11B. We observed very low levels of expression of these macrophage markers in KLN205 tumors irrespective of TMPRSS11B expression. We are cautious not to overinterpret these results and speculate that the discrepancy between infiltrating immune cells in the KLN205 and SNL models may be due to several factors, including differences in genetic background, the specific genetically engineered modifications, and the location of the tumors that develop *in vivo*. The SNL model is maintained on a mixed genetic background, is driven by Cre-mediated overexpression of Sox2 and loss of the tumor suppressors *Lkb1* and *Nkx2-1*, and autochthonous lung tumors develop spontaneously with mixed histology. In contrast, the KLN205 model was derived from a chemically-induced squamous tumor on the murine DBA/2 genetic background and transplanted subcutaneously in our study. These differences may contribute to distinct tumor microenvironments and lower levels of macrophage infiltration/polarization in the KLN205 syngeneic model.

3) It would be nice, at least in vitro, to demonstrate the reduction of TMPRSS11B protein levels upon Dox-induced shRNA expression. At the same time, additional controls should be included = shRNA1 and shRNA2 without Dox to compare to control shRNA with or without Dox, to show that the system is not leaky.

We thank the reviewer for this comment. As requested, we incorporated the additional controls, including shRNA1 and shRNA2 without doxycycline (dox) compared to control shRNA with or without dox. These data (provided below and in **new Fig. EV1C**) show that *Tmprss11b* expression is reduced with shRNA1 and shRNA2 only in the presence of dox treatment and not without dox treatment. Additionally, control shRNA does not significantly reduce *Tmprss11b* expression with or without dox treatment. These additional controls strengthen our overall conclusions that *Tmprss11b* depletion reduces tumor growth *in vivo* and confirm that our results are not due to leakiness of the dox-mediated shRNA system.

It is important to note that we previously tested several commercially available TMPRSS11B antibodies, but these failed to recognize endogenous TMPRSS11B. We also attempted to generate our own TMPRSS11B antibodies with New England Peptide without success (perhaps due to cross reactivity with closely related TMPRSS11 family members). To circumvent this issue, we utilize quantitative real-time qPCR to detect the reduction in *Tmprss11b* expression. Importantly, we utilized multiple Taqman probes targeting different regions of the *Tmprss11b* transcript in all experiments.

4) Fig. 3B and 4A. Region S3 seems to display intermediate TMPRSS11B expression. Could this be a tumor with an adeno-to-squamous transition? To investigate this, a staining with KRT16 and LYPD3 could help, as performed in Fig. S2F.

We thank the reviewer for raising this question, as we also considered the possibility that Region S3 with intermediate *Tmprss11b* expression may be a tumor undergoing an adeno-to-squamous transition. Histological analysis was performed in consultation with Dr. Bret Evers, a pathologist and co-author of this study. He confirmed that Region S3 is consistent with that of a lung squamous cell carcinoma and does not appear to be an adeno-squamous tumor (based on H&E staining). As requested, we assessed *Krt16* and *Lypd3* expression in these regions from our spatial dataset. Interestingly, *Krt16* is significantly enriched in the S1 & S2 Regions compared to the S3 Region, while *Lypd3* is expressed at similar levels. These data are provided below for the editor and reviewers:

We also performed IHCs for KRT16 and LYPD3 on FFPE serial sections from the same block of tissue used for spatial transcriptomics. Unfortunately, due to the distance between the serial sections used for spatial transcriptomics and the subsequent IHC analyses, we found that Regions S1, S2 and S3 changed considerably with respect to size and histology. This precluded us from making definitive conclusions correlating KRT16 and LYPD3 protein expression with high (S1/S2) versus intermediate (S3) *Tmprss11b* levels. Based on the histological assessment and the spatial data provided above, we cannot conclude that Region S3 is undergoing an

adeno-to-squamous transition. Regardless, the IHC staining provided in **Fig. EV2F** accurately reflects overall higher levels of KRT16 and LYPD3 staining in LUSCs compared to LUADS.

Although outside the scope of this paper, our spatial transcriptomics data allowed us to assess transcriptionally distinct Leiden clusters and infer copy number variation (CNV) profiles within the tissue and tumor subtypes. This analysis, provided below for the reviewers and editor, reveals that Regions S1 and S2 (with high *Tmprss11b*, corresponding to #3 in red) cluster separately from Region S3 (with intermediate *Tmprss11b*, corresponding to #6 in pink). Collectively, our data suggest that differences in gene expression and inferred CNV patterns may contribute to the clustering of distinct tumor regions and/or histotypes. Future work, including more direct genetic analyses and lineage tracing, will be aimed at addressing this.

Figure for referee with unpublished data has been removed upon request by the authors.

Referee #2:

1. Does this manuscript report a single key finding?

YES - LUSC tumors with elevated *TMPRSS11B*, a trypsin-like serine protease that promotes MCT4-mediated lactate export, exhibit a tumor microenvironment depleted of CD4/CD8 T cells and enriched with immunosuppressive M2-like macrophages.

2. Is the reported work of significance (YES), or does it describe a confirmatory finding or one that has already been documented using other methods or in other organisms etc (NO)?

YES - This study investigates the role of *TMPRSS11B* in shaping the tumor microenvironment (TME) of LUSC tumors using immunocompetent mouse models, an area not previously published.

3. Is it of general interest to the molecular biology community? YES/NO

If YES, please say why, in a single sentence. If NO, please state which more specialized community you feel it is aimed at (or none), in a single word or phrase.

This would be of greater interest to the cancer research community.

4. Is the single major finding robustly documented using independent lines of experimental evidence (YES), or is it really just a preliminary report requiring significant further data to become convincing, and thus more suited to a longer-format article (NO)? YES - The authors present a manuscript examining the role of *Tmprss11b* in regulating the tumor microenvironment (TME) in lung squamous cell carcinoma (LUSC), a subtype of NSCLC with limited effective therapeutic options. This study employs several immunocompetent mouse models of LUSC to demonstrate that: (a) loss of *Tmprss11b* slows tumor growth and increases CD4/CD8 T-cell infiltration; (b) spatial transcriptomics reveals the highest *Tmprss11b* expression in *Krt13+* hillock-like cells; (c) spatial transcriptomics indicates enrichment of M2 macrophages in *Tmprss11b*-expressing LUSC tumors; and (d) metabolomics data show elevated lactate levels in LUSC, with higher lactate in areas of lower pH compared to areas with higher pH. Overall, this well-written manuscript effectively leverages immunocompetent mouse models of LUSC and spatial transcriptomics to characterize the TME in *Tmprss11b*-high LUSC, *Tmprss11b*-low LUSC, and LUAD tumors.

Major Comments:

1) Figure 6C shows the accumulation of the pH-sensitive probe (USP 5.3 ICG). The authors argue that ICG accumulates in regions adjacent to *Tmprss11b*-high LUSC tumors; however, Figure 6C shows ICG-positive areas that are not directly adjacent to *Tmprss11b*-high LUSC tumors, with ICG accumulating in patches rather than surrounding *Tmprss11b*-high LUSC tumors. How do the authors reconcile this discrepancy? Could the use of the MCT1 inhibitor AZD3965 or sodium lactate, as described in reference 70, help address this question?

We acknowledge that the pH sensitive probe sometimes accumulates in patches rather than explicitly surrounding the *Tmprss11b*-high LUSC tumors. These patches are located in tumor stromal areas, which is consistent with our prior data showing the highest level of acidity at the tumor and stromal interface, where cancer cells secrete lactic acid into the stromal areas, leading to the nanoprobe activation and internalization by stromal cells (Feng *et al.* 2024). An independent study (in preparation) using head and neck cancer patient samples revealed nanoprobe enrichment in areas with high vascular density which serves as the entry point for the nanoparticles. To investigate this in our lung cancer model, we analyzed vasculature gene signatures (generated using our spatial data) and observed a similar degree of vascularization around the S2 (*Tmprss11b*-high LUSC) and A1 (LUAD) regions compared to the other tumor regions (S1, S3, A2, shown below). Interestingly, we observed stronger overlap between the ICG nanoprobe signal and the vascular gene score around S2 compared to A1, suggesting increased acidification around LUSCs compared to LUADs when comparing regions with a similar degree of vascularization. This is further supported by the higher glycolytic gene signatures in *Tmprss11b*-high LUSCs (see our response to **point #5** below from this same reviewer and **new Fig. EV9A-B**). Our collaborator, Dr. Jinming Gao, is currently investigating the link between tumor vasculature and ICG nanoprobe accumulation in different tumor types and we therefore prefer to publish these findings in a separate study.

Figure for referee with unpublished data has been removed upon request by the authors.

We also agree with the reviewer that the use of the MCT1 inhibitor AZD3965, as described in Feng *et al.* 2024, might be an effective approach to assess the connection between the *Tmprss11b-high* LUSCs and ICG regions. We would like to note that it would take at least an additional year to perform these experiments in SNL mice, due to the time needed to breed animals, infect with Adeno-Cre, and age animals for up to 8 months. Given the time constraints, we hope the editor and reviewers agree that this is beyond the scope of the current study.

2) Figure 7D compares lactate levels in high-pH versus low-pH areas, but based on Figure S7, it appears that the "high-pH" area primarily consists of normal tissue, while the "low-pH" area is tumor tissue (unspecified as LUSC or LUAD). Can the authors confirm if this interpretation is correct? Similarly, for Figures 6D and 6E, how were the "low-pH" and "high-pH" areas defined?

We thank the reviewer for this suggestion. We performed additional analysis and confirmed that the reviewer's interpretation is correct. The "low pH" regions do have more overlap with tumor regions (LUSC, LUAD or Adenosquamous) than the "high pH" regions. We have added this data to the manuscript as **new Fig. EV8B on p.8**.

Regarding **Figures 6D-E**, the "low pH" areas are defined as the regions with the highest accumulation of the ICG signal and the "high pH" regions are defined as the rest of the area excluding the "low pH" regions in the tissue section used for the spatial transcriptomics. We have updated the Materials and Methods section on **p.17-18** of the revised manuscript to reflect this.

3) Figures 5B, S5B, S5C, and S6D present quantification of various immune cell populations, including "leukocytes," which are enriched in *Tmprss11b-high* LUSC. Are "leukocytes" defined as CD45-positive cells? It would be helpful to validate some of the transcriptomics results for immune cells using IHC staining.

We performed immune cell deconvolution analysis using Tangram (as described in Cable DM *et al.*, *Nature Biotechnology*, 2022) due to its ability to map single cells onto spatial transcriptomics

data across multiple platforms with high accuracy. We added a **new Table EV3** listing the top 20 differentially expressed genes for each immune cell type utilized by the Tangram program for immune cell deconvolution. The “leukocytes” were defined using the 20 genes listed in **new Table EV3**. Although *Ptprc* (CD45) was not included in Tangram analysis, we assessed *Ptprc* expression in the LUSCs and LUADs and found that it correlates with leukocyte abundance in these tumor regions. These data are provided below.

To validate several of the immune cell types and in response to **Point #4** from this same reviewer, we performed additional IHCs for several neutrophil markers including MPO, CD11b, and LY6G (provided in **new Fig. EV5C**). In addition, we validated several macrophage markers (including MSR1, HMOX1, and ARG1: shown in **Figure 5E** and **Figure 6F**). Given that our main findings focus on macrophages and neutrophils, we decided to focus our IHC validation on these immune cells. We hope the reviewers and editor agree that it is not necessary to exhaustively perform IHC validation for additional immune cell subsets that are not central to our main conclusions.

4) Neutrophils have been reported to be overrepresented in LUSC compared to LUAD patient samples (PMID: 36368318), yet neutrophils are not present in the Figure S5B, S5C, S6D. It would be helpful to include neutrophils in this dataset.

The single cell RNA-sequencing dataset that was initially used as the reference for the immune cell deconvolution analysis lacked signatures corresponding to neutrophils, which explains why neutrophils were not represented in the **original Figure S5B-C** and **Figure S6D**. To address the reviewer’s comment, we performed additional gene set enrichment analyses using the WebGestalt (WEB-based Gene Set Analysis Toolkit). Importantly, we observed significant enrichment for neutrophils in the *Tmprss11b*-high LUSCs when compared to *Tmprss11b*-low LUSC and LUADs. These data are provided in new **Fig. EV5A-B** and are consistent with the

findings in the LUSC patient samples from PMID: 36368318 (Salcher et al, *Cancer Cell*, 2022) and PMID: 28146145 (Kargl et al, *Nat Commun*, 2017). We also performed IHC analysis for several neutrophil markers, including MPO, CD11b, and LY6G. These data, provided in **new Fig. EV5C**, demonstrate that neutrophil markers are indeed expressed at higher levels in squamous tumors compared to lung adenocarcinomas, consistent with the publications above and Dr. Oliver's prior publication describing the SNL mouse model, PMID: 30332632 (Mollaoglu et al, *Immunity*, 2018).

5) Do *Tmprss11b*-high LUSC tumors exhibit elevated levels of *Glut1*, *MCT1*, and *MCT4*?

We thank the reviewer for this suggestion. To address this, we first assessed glycolytic and TCA cycle gene signatures in our spatial data and observed significant enrichment for the glycolytic signature but not the TCA cycle signature in *Tmprss11b*-high LUSCs. These data are provided in **new Fig. EV9A-B**. We also added a **new Table EV2** listing the gene signatures, and a description of the analysis has been added to the Materials and Methods section on **p.17-18** of the revised manuscript.

We also performed multiplex fluorescent IHC (IHC-F) for GLUT1, MCT1, and MCT4, as requested by this reviewer. Interestingly, we observed higher levels of GLUT1 and MCT4 in the LUSCs, but not MCT1 (**new Fig. EV9C**). This is consistent with the observation that squamous tumors generally have more prominent glycolytic metabolism compared to adenocarcinomas, and prior reports suggesting GLUT1 is enriched in LUSC compared to LADC (PMID: 28548087: Goodwin et al, *Nat Commun*, 2017). MCT1 and MCT4 are known to be expressed heterogeneously in human lung tumors, and we hope the reviewer and editors will agree that dissecting this heterogeneity in SNL tumors is beyond the scope of the current study.

Minor comments:

1) Some of author affiliations appear (possibly) incorrect.

Thank you for noticing this. We confirm that all author affiliations are correct in the revised manuscript.

2) Figure 3E, 5D, 6E, S2E, S5E - I would suggest re-naming this to "Genes enriched in..."

We have updated the text and figure legends, accordingly.

3) Line 540 - "24-hours post-infection" should be "24-hours post-injection"?

Thank you for noticing this. We updated this to read "24-hours post-injection".

4) Supplementary Table 1 lists PDL1, FOXP3, CD25 antibodies that were used in IHC, but there are no Figures that include PDL1, FOXP3, CD25.

We updated the Supplementary Table and removed PD-L1, FOXP3 and CD25. Please note that all antibody information is now included in the **Reagents and Tools Table**, which is part of the Structured Methods in the revised manuscript.

5) Supplementary Table S4 lists metabolites identified (16 total), but only lactate was included in the Figures.

We did perform similar analyses for several other metabolites that were initially included in **Supplementary Table S4** (renamed **Table EV4**). Interestingly, we observed significantly higher levels of glutamic acid in LUSCs and LUADs compared to normal tissues. Additionally, a similar increase in glutamic acid was seen in low pH regions compared to high pH regions. In contrast, we observed significantly lower levels of succinic acid in LUSCs and LUADs compared to normal tissues. While these data are very interesting and merit follow up studies, we do not want to distract from our central findings that *Tmprss11b*-high squamous tumors and acidified regions have elevated levels of lactate. We therefore decided to remove the other metabolites from **Table EV4**. Although beyond the scope of this paper, future work may focus on investigating glutamic acid and succinic acid in lung squamous tumors using detailed metabolic analysis.

Dear Dr. O'Donnell

Thank you for the submission of your revised manuscript to our editorial offices. I have now received the reports from the two referees that I asked to re-evaluate the study, you will find below. As you will see, both referees now fully support the publication of your study in EMBO reports.

Before we can proceed with formal acceptance, I have the following editorial requests I ask you to address in a final revised manuscript:

- Please provide a final comprehensive title with not more than 100 characters (including spaces).
- There are author name discrepancies. It is William Harnett in the manuscript, but William Hartnett in the submission system, and William T. Putnam in the manuscript and William T. Putman in the submission system. Please check.
- Please provide the abstract written in present tense throughout.
- Please reduce the number of keywords to 5.
- Please reduce the number of EV figures. Usually, EMBO reports articles have not more than 6 EV figures. Please re-arrange the figures in order to have not more than 6 final EV figures. Please update all the callouts and re-arrange the source data files accordingly.
- Please remove now the referee tokens from the data availability section and make sure that all datasets are public latest upon publication of the manuscript.
- Please check again that the number "n" for how many independent experiments were performed, their nature (biological versus technical replicates), the bars and error bars (e.g. SEM, SD) and the test used to calculate p-values is indicated in the respective figure legends. Please also check that all the p-values are explained in the legend, and that these fit to those shown in the figure. Please provide statistical testing where applicable. Please avoid the phrase 'independent experiment' but clearly state if these were biological or technical replicates. Please also indicate (e.g. with n.s.) if testing was performed, but the differences are not significant. In case n=2, please show the data as separate datapoints without error bars and statistics (please check for panels 1I, and EV5B/D). See also:
<http://www.embopress.org/page/journal/14693178/authorguide#statisticalanalysis>

If n<5, please show single datapoints for diagrams. Moreover:

- Please define the annotated p values ****/**/*/* as well as provide the exact p-values for the same in the legend of figure 2A, 5B, EV6 A, B; EV7 D; EV9 B as appropriate.
- Please note that the exact p values are not provided in the legends of figures 1B, E; 4C, D; EV1 C, EV4 B
- Please indicate the statistical test used for data analysis in the legends of figures 2A, 3D, E; 5B, C; 6D, E, EV2 B, D, E; EV4 B; EV6 A, B, C, D; EV7 C, D; EV9 B"
- Please note that the box plots need to be defined in terms of minima, maxima, centre, bounds of box and whiskers, and percentile in the legends of figures 5B, EV6 A, B; EV7 D
- Please note that information related to n is missing in the legends of figures 2A, 5B, EV2 B, C; EV4 B; EV6 A, B; EV 7 D, EV9 B
- Please note that the error bars are not defined in the legend of figure 2A.
- Please add the information provided in Table EV1 to the Reagents & Tools table and remove Table EV1 from the manuscript files. Please update all callouts.
- Table EV2 is a dataset. Please upload the original Excel file as a dataset files (named 'Dataset EV1'). Please add a legend on the first TAB of the Excel file and update all callouts.
- Please update the numbering of the remaining EV tables and update their callouts.
- It seems the lung section shown in Fig. 3A is shown also in Figs. 3B/C, 4A, 6C, EV4A, EV5B and EV9A. Please make sure this re-use is explained and clearly indicated in the respective figure legends.

In addition, I would need from you uploaded separately:

- a short, two-sentence summary of the manuscript (not more than 35 words).
- two to four short (!) bullet points highlighting the key findings of your study (two lines each).
- a schematic summary figure as separate file that provides a sketch of the major findings (not a data image) in jpeg or tiff format (with the exact width of 550 pixels and a height of not more than 400 pixels) that can be used as a visual synopsis on our

website.

Best,

Referee #1:

The authors have satisfactorily responded to my comments.

Referee #2:

The authors have comprehensively resolved all concerns I raised in my review, and I have no additional questions.

All editorial and formatting issues were resolved by the authors.

Dr. Kathryn O'Donnell
University of Texas Southwestern Medical Center
Molecular Biology
United States

Dear Dr. O'Donnell,

I am very pleased to accept your manuscript for publication in the next available issue of EMBO reports. Thank you for your contribution to our journal.

Yours sincerely,
